# Heavy water inhibits DNA double-strand break repairs and disturbs cellular transcription, presumably via quantum-level mechanisms of kinetic isotope effects on hydrolytic enzyme reactions

Takeshi Yasuda[1]*, Nakako Nakajima[2], Tomoo Ogi[3], Tomoko Yanaka[1], Izumi Tanaka[4], Takaya Gotoh[5], Wataru Kagawa[6], Kaoru Sugasawa[7], Katsushi Tajima[8]

1 Institute for Quantum Life Science, National Institutes for Quantum Science and Technology, Chiba, Japan, 2 QST Hospital, National Institutes for Quantum Science and Technology, Chiba, Japan, 3 Department of Genetics, Research Institute of Environmental Medicine, Nagoya University, Nagoya, Japan, 4 Institute for Radiological Sciences, National Institutes for Quantum Science and Technology, Chiba, Japan, 5 Department of Health Science, Daito Bunka University, Saitama, Japan, 6 Department of Interdisciplinary Science and Engineering, Program in Chemistry and Life Science, School of Science and Engineering, Meisei University, Tokyo, Japan, 7 Biosignal Research Center, and Graduate School of Science, Kobe University, Kobe, Japan, 8 Department of Hematology, Yamagata Prefectural Central Hospital, Yamagata, Japan

* yasuda.takeshi@qst.go.jp

## Abstract

Heavy water, containing the heavy hydrogen isotope, is toxic to cells, although the underlying mechanism remains incompletely understood. In addition, certain enzymatic proton transfer reactions exhibit kinetic isotope effects attributed to hydrogen isotopes and their temperature dependencies, indicative of quantum tunneling phenomena. However, the correlation between the biological effects of heavy water and the kinetic isotope effects mediated by hydrogen isotopes remains elusive. In this study, we elucidated the kinetic isotope effects arising from hydrogen isotopes of water and their temperature dependencies *in vitro*, focusing on deacetylation, DNA cleavage, and protein cleavage, which are crucial enzymatic reactions mediated by hydrolysis. Intriguingly, the intracellular isotope effects of heavy water, related to the *in vitro* kinetic isotope effects, significantly impeded multiple DNA double-strand break repair mechanisms crucial for cell survival. Additionally, heavy water exposure enhanced histone acetylation and associated transcriptional activation in cells, consistent with the *in vitro* kinetic isotope effects observed in histone deacetylation reactions. Moreover, as observed for the *in vitro* kinetic isotope effects, the cytotoxic effect on cell proliferation induced by heavy water exhibited temperature-dependency. These findings reveal the substantial impact of heavy water-induced isotope effects on cellular functions governed by hydrolytic enzymatic reactions, potentially mediated by quantum-level mechanisms underlying kinetic isotope effects.

**Data Availability Statement:** All relevant data are within the paper and its Supporting Information files.

**Funding:** This study was funded by the Japan Society for the Promotion of Science (JSPS) KAKENHI grants 20K12177 (to TYas), 20K08071 (to KT and TYas), 22H03743 (to WK and TYas), and 22K08414 (to TG and TYas). This study was also funded by the joint research program of the Biosignal Research Center, Kobe University, grants 281004 (to TYas and KS), and 201003 (to TYas and KS). The funders had no role in the study design, data collection, interpretation, or decision to submit the work for publication.

**Competing interests:** The authors have declared that no competing interests exist.

## Introduction

Hydrolysis is a chemical reaction in which a water molecule directly participates in breaking the chemical bonds of a reactant, resulting in decomposition products [1, 2]. One proton of $H_2O$ is transferred to a reactant during the hydrolysis reaction. Living organisms have huge numbers of hydrolytic enzymes that play a variety of roles by catalyzing hydrolysis reactions. For example, nucleases, which cleave DNA and RNA, are involved in various nucleic acid metabolic reactions such as DNA repair and DNA replication [3–7]. In addition, proteases, which cleave peptide bonds of proteins by hydrolysis, degrade or regulate proteins by specific cleavage within the molecules [8–12]. The activities and functions of many proteins in cells are regulated by modifications such as phosphorylation [13, 14], and phosphatases, enzymes that remove phosphorylation modifications, are also hydrolases [15]. Glycosidases, which are involved in the degradation of polysaccharides, are hydrolases [16], as are ATPases, which are involved in the production of energy in the body [2, 17, 18]. Thus, water molecules in cells not only serve as a solvent, but also are directly involved in a wide range of enzymatic chemical reactions.

Like the hydrolytic enzymes, dehydrogenase family proteins and thymidylate synthase also catalyze proton transfer reactions, and previous studies showed that quantum mechanisms are involved in these reactions [19–24]. A quantum is a very small, discrete unit of matter or energy, such as a proton, electron, or photon, with both particle and wave properties. Therefore, enzyme-catalyzed chemical reactions involving a proton, electron, or photon are expected to behave quantum mechanically. In a chemical reaction, the starting reactant exists in the lowest energy state, called the ground state (GS). For the chemical reaction to proceed, it must overcome an activation energy barrier, in which the highest energy state is the transition state (TS). The activation energy barrier can be surmounted by heat in a chemical reaction. A catalyst promotes a chemical reaction without heating, by forming a reaction intermediate complexed with the starting material, and thereby provides a new reaction pathway with a lower activation energy barrier. In quantum phenomena, quantum particles can pass through the barrier, allowing the chemical reaction to proceed without reaching the transition state. This phenomenon is called "quantum tunneling". Quantum tunneling occurs in reactions involving electrons and hydrogen, with small atomic weights, but not atoms with larger atomic weights [25, 26]. Therefore, the probability of quantum tunneling is higher for hydrogen (H), with an atomic weight of 1, than for deuterium (D), an isotope of hydrogen with an atomic weight of 2, resulting in a faster chemical reaction rate for H than for D. The change in reaction rate that occurs when one of the atoms of a reactant is substituted by an isotope is called the kinetic isotope effect. Thus, the kinetic isotope effect due to H and D is related to the quantum tunneling effect. By using the kinetic isotope effect as a probe, the quantum tunneling effect was examined in hydrogen-transfer reactions by dehydrogenase family proteins and thymidylate synthase [19–24]. In these reactions, the kinetic isotope effect was examined by comparing the transfer of normal hydrogen ($^1H$ or H) and its heavier isotope deuterium ($^2H$ or D) or tritium ($^3H$ or T) [19–24]. The transfer rate of normal hydrogen by the enzymes was faster than those of deuterium and tritium, indicating that the quantum tunneling effect occurs in these enzyme reactions.

Although the quantum tunneling effects in dehydrogenase family proteins and thymidylate synthase have been reported, the extent of these effects in various types of reactions by hydrolytic enzymes that catalyze proton-transfer from a water molecule remains enigmatic. In addition, the biological effects derived from the quantum tunneling have not been clarified [27]. Meanwhile, deuterated water, in which the hydrogen of a water molecule is replaced by deuterium, is reportedly toxic to cells and animals [28–35]. Exposure to heavy water caused

inhibition of cell division and induction of cell death [30, 34, 35], but the mechanism was not fully elucidated. Therefore, this study aimed to determine the kinetic isotope effects due to heavy water on several types of hydrolytic enzymes related to various important biological phenomena and their associated effects on cells.

## Materials and methods

### Recombinant proteins and reagents

The recombinant human p300 and RAD52 (full length) proteins were purified as described previously [36, 37]. Commercially purchased recombinant SIRT3 protein (#50014, Lot 2003, BPS Bioscience, San Diego, CA, USA) was used as described previously [37]. Recombinant human histone H2A/H2B/H3.1/H4 octamer proteins were kindly provided by Dr. H. Kurumizaka [38]. The following reagents were purchased: Acetyl Coenzyme A (A2181, Merck & Co., Inc., Whitehouse Station, NJ, USA), $^{14}$C-labeled Ac-CoA (ARC0554, Muromachi Kikai Co., Ltd., Tokyo, Japan), NAD$^+$ (N7004, Merck & Co., Inc.), BSA (B9001, New England Biolabs, Ipswich, MA, USA), I-SceI (R0694, New England Biolabs), Caspase 3 active (14–264, Merck & Co., Inc.), H$_2$O (Fig 1B and 1C, and #1 in S2 Fig; purified with a Milli-Q Synthesis A10 water purification system, Merck & Co., Inc.), H$_2$O (#2 in S2 Fig, and other figures unless otherwise noted; 06442–95, Nacalai Tesque, Inc., Kyoto, Japan), H$_2$O (#3 in S2 Fig; H20MB0501, Merck & Co., Inc.), D$_2$O (Fig 1B and 1C, and #1 in S2 Fig; 151882, Merck & Co., Inc.), D$_2$O (#2 in S2 Fig, and other figures unless otherwise noted; D214H, Nacalai Tesque, Inc.).

### Immunoblotting

Immunoblotting analyses were performed as described previously [37].

### Antibodies

The following antibodies were used: anti-acetyl lysine (#9441, Cell Signaling, Danvers, MA, USA), anti-RAD52 (sc-8350, Santa Cruz Biotechnology, Dallas, TX, USA), anti-acetyl Histone H3 (Lys9) (06–942, Merck & Co., Inc.), anti-Histone H3 (ab1791, abcam, Cambridge, UK), anti-phospho-ATM at Ser1981 (#4526, Cell Signaling), anti-53BP1 (#4937, Cell Signaling), anti-phospho-BRCA1 at Ser1524 (#9009, Cell Signaling), anti-RAD51 (70–001, Bio Academia, Osaka, Japan), anti-γH2AX clone JBW301 (05–636, Merck Millipore) and anti-GAPDH (#2118, Cell Signaling).

### Microscopy

The immunofluorescence microscopic analysis of the cells irradiated with γ rays was performed with anti-phospho-ATM at Ser1981, anti-53BP1, anti-phospho-BRCA1 at Ser1254, anti-RAD51, and anti-γH2AX antibodies, using an IX70 fluorescence microscope (Olympus, Tokyo, Japan) equipped with an ORCA-R2 cooled CCD camera (Hamamatsu Photonics, Hamamatsu, Japan) and x100 lens (UPLSAPO100XO, Olympus), as described previously [37]. For the quantitative colocalization analysis, the total number of phospho-ATM, 53BP1, phospho-BRCA1, or RAD51 foci and the number of their foci colocalized with γH2AX were counted in each cell. The percentage of colocalized foci was calculated by dividing the number of colocalized foci by the total number of foci.

Phase-contrast images were obtained with an all-in-one fluorescence microscope (BZ-X700, Keyence, Osaka, Japan) equipped with a x20 lens.

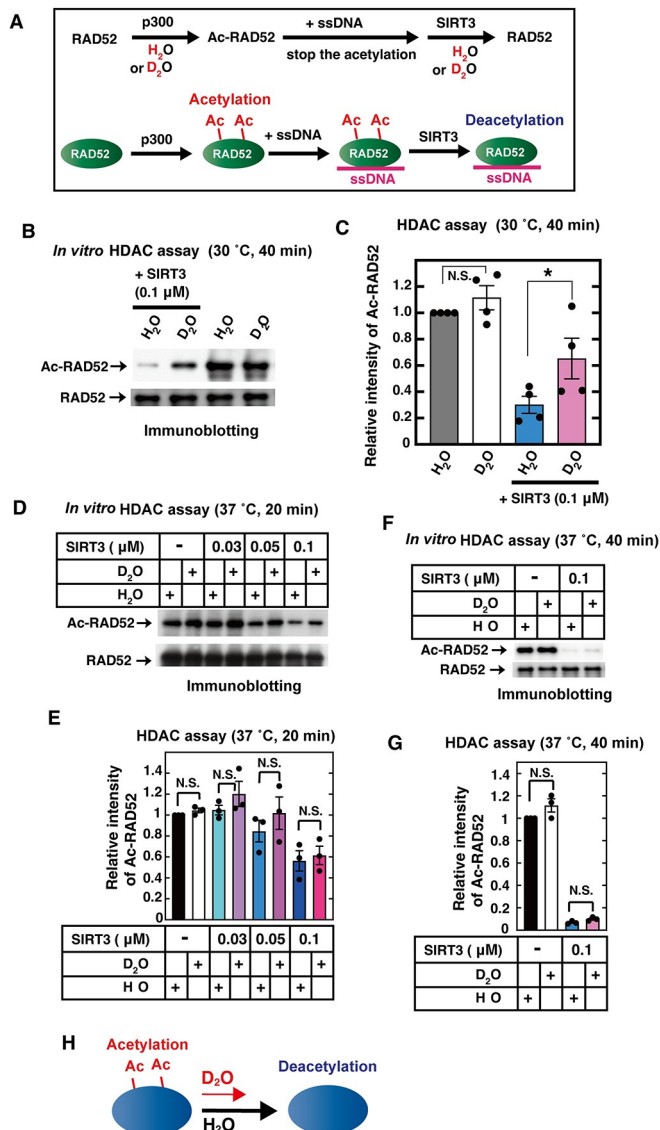

**Fig 1. Kinetic isotope effect on RAD52 protein deacetylation by SIRT3.** (**A**) Schematic representation of the RAD52 deacetylation assay. (**B**, **D**, and **F**) *In vitro* acetylation assays of RAD52 were performed, as described in the Materials and Methods. After the addition of a poly dT 68 mer, aliquots of the reaction mixture containing RAD52 proteins (0.1 μM in Fig 1B and 0.08 μM in Fig 1D and Fig 1F) were incubated with the indicated amount of SIRT3 in HDAC buffer in the presence of $H_2O$ or $D_2O$, at the indicated temperature for the indicated times. The reaction mixtures were subjected to SDS-PAGE, followed by immunoblotting with an anti-acetylated lysine antibody (top: Ac-RAD52) and an anti-RAD52 antibody (bottom: RAD52). (**C**, **E**, and **G**) Relative band intensities of acetylated RAD52 normalized to those of the RAD52 bands. The graph shows the mean values and standard errors of the mean from 4 (C) or 3 (E and G) independent experiments, with dots of each data values. The samples connected by lines were compared (*$P < 0.05$ and N.S., not significant by an unpaired Student's t-test). (**H**) Schematic representation of the experimental results for the kinetic isotope effect on the protein deacetylation reaction.

## *In vitro* acetylation and deacetylation assays

*In vitro* acetylation assays of RAD52 were performed by incubating p300 (0.13 μM) and RAD52 (1.7 μM) proteins in HAT buffer [50 mM Tris-HCl, 1 mM EDTA, 10% glycerol, 1 mM DTT, pH 8.0], made with $H_2O$ (06442–95, Nacalai Tesque, Inc.) or $D_2O$ (D214H, Nacalai Tesque, Inc.), in the presence of Ac-CoA (100 μM) at 30°C for 60 min. Subsequently, a poly dT 68

mer (14.8 μM; synthesized by Operon Biotechnologies, Japan) was added to the reaction mixture, which was further incubated at 30˚C for 10 min to inhibit the acetylation reaction of RAD52, as described [37]. *In vitro* deacetylation assays of the acetylated RAD52 proteins were then performed by incubating an aliquot of the reaction mixture containing the acetylated RAD52 (0.1 or 0.08 μM) with the indicated amount of recombinant SIRT3 protein in HDAC buffer [25 mM Tris-HCl, 137 mM NaCl, 2.7 mM KCl, 1 mM MgCl$_2$, 0.1 mg/ml BSA, pH 8.0], made with H$_2$O (06442–95, Nacalai Tesque, Inc.) or D$_2$O (D214H, Nacalai Tesque, Inc.), in the presence of 500 μM NAD$^+$ at the indicated temperature. The reaction mixtures were subjected to SDS-PAGE, followed by immunoblotting with an anti-acetylated lysine antibody and an anti-RAD52 antibody.

*In vitro* acetylation assays of histone proteins were performed by incubating p300 (0.567 μM) and human histone H2A/H2B/H3.1/H4 octamer proteins (4.62 μM) in HAT buffer [50 mM Tris-HCl, 1 mM EDTA, 10% glycerol, 1 mM DTT, pH 8.0], made with H$_2$O (06442–95, Nacalai Tesque Inc.) or D$_2$O (D214H, Nacalai Tesque Inc.), in the presence of $^{14}$C-labeled Ac-CoA (170 μM) at 30˚C for 60 min. Subsequently, *in vitro* deacetylation assays of the acetylated histone proteins were performed by incubating an aliquot of the reaction mixture containing the acetylated histone octamer proteins (0.67 μM) with the indicated amount of recombinant SIRT3 protein in HDAC buffer [25 mM Tris-HCl, 137 mM NaCl, 2.7 mM KCl, 1 mM MgCl$_2$, 0.1 mg/ml BSA, pH 8.0], made with H$_2$O (06442–95, Nacalai Tesque Inc.) or D$_2$O (D214H, Nacalai Tesque Inc.), in the presence of 500 μM NAD$^+$ at the indicated temperature. The reactions were fractionated by SDS-PAGE, and the gels were stained with Coomassie Brilliant Blue. The dried gels were photographed using an EOS 6D digital camera (Cannon, Tokyo, Japan) equipped with an EF 24–70 mm F4L IS USM lens (Cannon), and exposed to an imaging plate (Fuji Film, Tokyo, Japan). Radioisotopic images in the exposed imaging plate were analyzed by an FLA-3000 fluorescent image analyzer (Fuji Film).

### *In vitro* caspase-3-mediated peptide cleavage assay

*In vitro* caspase-3-mediated DEVD peptide cleavage assays were essentially performed as described previously [39], using a Caspase 3/7 Assay Kit (17–367, Merck & Co., Inc.). The active caspase-3 recombinant protein (25 ng) (14–264, Merck & Co., Inc.) was incubated at the indicated temperature with the Ac-DEVD-AMC substrate solution (0.75 μl) (12–541, Merck & Co., Inc.) in 75 μl of 1.5 x Caspase buffer [30 mM PIPES, pH 7.2, 112.5 mM NaCl, 3.75 mM EDTA, 0.15% CHAPS, 11.25% sucrose, 15 mmol DTT], made with H$_2$O (06442–95, Nacalai Tesque, Inc.) or D$_2$O (D214H, Nacalai Tesque, Inc.). Following the addition of the Ac-DEVD-AMC substrate, the fluorescence intensity resulting from substrate cleavage was measured with excitation at 355 nm and emission at 460 nm, using Wallac 1420 ARVOsx and ARVO X5 microplate readers (Perkin Elmer, Waltham, MA, USA).

### *In vitro* I-SceI-mediated DNA cleavage assay

For the *in vitro* I-SceI-mediated DNA cleavage assay, the pGP2 NotI-linearized control plasmid (New England Biolabs), which contains a single I-SceI site, was used as the substrate. The substrate DNA (50 ng) was incubated with the indicated amount of I-SceI enzyme in 10 μl of CutSmart buffer [50 mM potassium acetate, 20 mM Tris-acetate, 10 mM magnesium acetate, 100 μg/ml BSA, pH 7.9], made with H$_2$O (06442–95, Nacalai Tesque, Inc.) or D$_2$O (D214H, Nacalai Tesque, Inc.), at the indicated temperature. After the reaction, the samples were analyzed by 0.8% agarose gel electrophoresis with ethidium bromide staining. The gel images were captured using a BioSpectrum Imaging System (UVP, Upland, CA, USA) and an LAS-

4000 mini imaging analyzer (Fujifilm, Tokyo, Japan). Quantification was performed using the Multi Gauge software (Fujifilm, Tokyo, Japan).

## Cell culture

HeLa pDR-GFP cells, obtained from Dr. M. Jasin [40, 41], as well as HEK293, HFL III, and U2OS cells, were cultured in minimum essential medium (MEM) (1030700, Thermo Fisher Scientific, Waltham, MA, USA), supplemented with 10% fetal bovine serum (FBS), 2 mM L-glutamine, and 1% penicillin-streptomycin (PS). For experiments comparing the kinetic isotope effects due to $H_2O$ or $D_2O$ in these cells, the MEM solutions were prepared by dissolving MEM powder (61100, Thermo Fisher Scientific) in $H_2O$ (06442–95, Nacalai Tesque, Inc.) or $D_2O$ (D214H, Nacalai Tesque, Inc.) according to the manufacturer's instructions.

U937, HCT116 (p53 wild type or null) and H1299 dA3-1#1 [42, 43] cells were cultured in RMPI 1640 medium (11875, Thermo Fisher Scientific) supplemented with 10% FBS and 1% PS. For experiments comparing the kinetic isotope effects due to $H_2O$ or $D_2O$ in the cells, the medium was prepared by dissolving RMPI 1640 powder (31800, Thermo Fisher Scientific) in $H_2O$ (06442–95, Nacalai Tesque Inc.) or $D_2O$ (D214H, Nacalai Tesque Inc.).

HEK293 cells for the SSA assay (HEK293 SAGFP cells) were constructed as described [44]. The hprtSAGFP plasmid (41594, addgene) was linearized with the KpnI and SacI restriction enzymes, and transfected into HEK293 cells with Lipofectamine 2000 (11668, Thermo Fisher Scientific). The transfected cells were cultured in DMEM (11965, Thermo Fisher Scientific) supplemented with 10% fetal bovine serum (FBS), 2 mM L-glutamine, and 1% penicillin-streptomycin (PS), and stable transfected cells were selected with 0.5 µg/ml of puromycin (A1138, Thermo Fisher Scientific) in the culture medium. For experiments comparing the kinetic isotope effects due to $H_2O$ or $D_2O$ in the cells, the medium was prepared by dissolving DMEM powder (12100, Thermo Fisher Scientific) in $H_2O$ (06442–95, Nacalai Tesque Inc.) or $D_2O$ (D214H, Nacalai Tesque Inc.).

## siRNA treatments

Stealth Select siRNAs were purchased from Thermo Fisher Scientific. Stealth Select siRNA, HSS118726 (5'-AAUCAGCUCAGCUACAUCCUGCAGG-3'), was used for the siRNA treatment against SIRT3. Stealth Select siRNA, HSS109021 (5'-GGCCAAUGAGAUGUUUGGUUACAAU-3'), was used for the siRNA treatment against RAD52. Stealth RNAi negative control siRNA oligonucleotides (Thermo Fisher Scientific) were used for the negative controls. Lipofectamine RNAiMAX was used for the transfection of Stealth Select siRNAs into cells, according to the transfection protocol (Thermo Fisher Scientific).

## I-SceI-based reporter assay for HR, SSA and NHEJ repairs

For the HR repair assay, Hela pDR-GFP cells ($5x10^5$ cells/well in a 12-well plate) were untreated or transfected with siRNA. At 24 h after the siRNA-transfection, the culture medium was changed to fresh medium made with $H_2O$ (06442–95, Nacalai Tesque, Inc.) or $D_2O$ (D214H, Nacalai Tesque, Inc.), and the cells were transfected with 0.5 or 2 µg of the I-SceI expression plasmid, pCMV-NLS-I-SceI [40, 41]. At 48 h after the plasmid-transfection, the cells were harvested by trypsinization and analyzed with a FACSverse flow cytometer (Becton Dickinson, San Jose, CA, USA).

For the SSA or NHEJ repair assay, HEK293 SAGFP or H1299 dA3-1#1 cells were seeded in a 12-well plate ($5x10^5$ cells/well) in DMEM or RMPI 1640 medium, respectively. The next day, the culture medium was changed to fresh medium made with $H_2O$ (06442–95, Nacalai Tesque, Inc.) or $D_2O$ (D214H, Nacalai Tesque, Inc.), and the cells were transfected with 2 µg of the

I-SceI expression plasmid, pCMV-NLS-I-SceI [40, 41]. At 48 h after the plasmid-transfection, the cells were harvested by trypsinization and analyzed with a SA3800 Spectral Cell Analyzer (SONY Biotechnology, San Jose, CA, USA).

GFP-positive cells were counted by using the FlowJo software (Tomy Digital Biology, Tokyo, Japan).

### RNA extraction and RNA-seq analysis

Hela pDR-GFP cells ($5x10^5$ cells/well in a 12-well plate) were untreated or transfected with siRNA. At 48 h after the siRNA-transfection, the cell culture medium was replaced with fresh medium. On the next day, the cell culture medium was changed to fresh culture medium made with $H_2O$ (06442–95, Nacalai Tesque Inc.) or $D_2O$ (D214H, Nacalai Tesque Inc.). After a 5 h incubation in the medium containing $H_2O$ or $D_2O$, the cells were harvested by trypsinization and total RNA was isolated, using an RNeasy Mini Kit (Qiagen) according to the manufacturer's instructions.

RNAseq was performed to investigate gene expression levels. mRNA was obtained from total RNA using the NEBNext Poly(A) mRNA Magnetic Isolation Module (NEB). The MGIEasy RNA Directional Library Prep Set (MGI Tech, Shinsen, China) was used to adjust the library. The libraries were sequenced on the DNBSEQ-G400 platform (MGI Tech), with paired-end flow cells to obtain 100–150 base pair reads with 100–200-fold coverage.

The obtained read counts data from RNA-seq experiments were analyzed with iDEP96 [45]. The heatmap was generated using the following criteria with iDEP96: distance–correlation, linkage–average and cut-off Z score–4. The KEGG pathway maps [46] were colored based on the results of KEGG Pathview rendering [47] by iDEP96. The fold-change (log2) cut-off in the color code is 2. Permission has been obtained from Kanehisa laboratories for using KEGG pathway map images.

### Cell survival assays

To monitor the cell viability by an MTT assay, cells were seeded in 96-well flat-bottom plates (3,000 cells/well) (167008, Thermo Fisher Scientific). After 24 h, the cell culture medium was changed to fresh medium composed of various ratios of $H_2O$ (06442–95, Nacalai Tesque, Inc.) and $D_2O$ (D214H, Nacalai Tesque, Inc.). After 6 days, the cell viability was examined with a ROCHE Cell Proliferation kit I (MTT) (11465007001, Merck & Co., Inc.), according to the manufacturer's instructions, using an ARVO X5 microplate reader (Perkin Elmer).

### Apoptosis and necrosis assays

For apoptosis and necrosis assays, cells were seeded in 96-well flat-bottom white plates (136101, Thermo Fisher Scientific) and 96-well flat-bottom black plates (137101, Thermo Fisher Scientific) at 10,000 cells/well. After 24 h, the cell culture medium was exchanged with fresh culture medium made with $H_2O$ (06442–95, Nacalai Tesque Inc.) or $D_2O$ (D214H, Nacalai Tesque Inc.). Apoptosis and necrosis assays were performed with a RealTime-Glo Annexin V Apoptosis and Necrosis Assay kit (JA1011, Promega, Madison, WI, USA) at the indicated times after changing the culture medium, using an ARVO X5 microplate reader (Perkin Elmer).

### Cell growth assay

Hela DR-GFP cells were seeded in 3.5 cm dishes (40,000 cells/dish). The cells were cultured at 31, 33, 35, 37, or 39°C in MEM culture medium containing 0, 20, or 50% $D_2O$. After 4 days,

the numbers of adherent cells were counted with a Z1 Coulter Counter (Beckman Coulter, Brea, CA, USA).

## Statistical analysis

The KaleidaGraph software, version 5.0, was used for statistical analyses. Statistical analyses for multiple comparisons were performed using a one-way ANOVA, as described previously [48]. Statistical analyses between the data of two groups were performed using an unpaired Student´s t-test, as described previously [48].

## Results

### Relationship between kinetic isotope effects and quantum-level mechanisms

The chemical reaction of the reactant mediated by quantum particles at the GS can proceed without reaching the TS by the quantum tunneling effect (S1A Fig). Because deuterium (D) has a lower probability of quantum tunneling than hydrogen (H), a kinetic isotope effect appears in enzymatic chemical reactions involving hydrogen, in which replacing H with D results in a lower reaction rate (S1B Fig, S1 Text). The connection between quantum tunneling and kinetic isotope effects can be understood from the equation for the probability of the quantum tunneling effect, which is derived from the Schrödinger equation (S1 Text) described in textbooks, such as that by P.W. Atkins [25, 26]. The probability of the quantum tunneling effect of D is lower than that of H, thereby resulting in the kinetic isotope effects. The kinetic isotope effect is also caused by differences in the vibrational potential energies of the chemical bonds involved in the reaction, which are also related to quantum theory (S1C Fig, S2 Text) [25, 26]. When the reaction temperature is high, the reaction can proceed beyond the potential energy peak without quantum tunneling. Therefore, the quantum tunneling effect is reduced or eliminated at higher temperatures, and accordingly, this temperature dependency is also an indicator [23–26]. In contrast, the isotope effects caused by the differences in the vibrational potential energy are less sensitive to temperature variation in the temperature range used in our experiments (S2 Text).

For example, a previous report on kinetic isotope effects on proton transfer in thymidylate synthase showed that in the temperature range of 5–35˚C, the change in the reaction rate of the enzyme reaction between D and T isotopes is about 2–1.5, while that between H and T isotopes is about 20–3 [22]. In addition, the kinetic isotope effect reportedly becomes smaller as the reaction temperature increases. The degree of the kinetic isotope effect in proton transfer depends on each enzymatic reaction [22, 49].

### Kinetic isotope effect on SIRT3-mediated human RAD52 deacetylation

Hydrogen-transfer is involved in deacetylation [50–52], which is the reverse reaction of acetylation. During deacetylation, a hydrogen of a water molecule directly attacks the chemical bond of acetyl-lysine to remove the acetyl group, and is transferred to the deacetylated lysine (S1D Fig). Acetylation is catalyzed by histone acetyltransferases (HATs), and deacetylation is accomplished by histone deacetylases (HDACs). Human RAD52, a DNA double strand break (DSB) repair protein [36], is acetylated by a HAT, p300, and is deacetylated by an HDAC, SIRT3 [37]. To examine whether the deacetylation reaction is subjected to the kinetic isotope effect, we performed *in vitro* RAD52 deacetylation by SIRT3 [37]. At first, RAD52 is incubated at 30˚C with p300 and Acetyl Coenzyme A (Ac-CoA) in reaction buffer containing $H_2O$ or $D_2O$, for the RAD52 acetylation (HAT assay). Because RAD52 acetylation is inhibited in the

presence of single stranded (ss)DNA [37], ssDNA is then added to stop the ongoing RAD52 acetylation reaction. A portion of the reaction mixture containing the acetylated RAD52 is incubated with SIRT3 and $NAD^+$ in buffer containing $H_2O$ or $D_2O$, at 30˚C (HDAC assay) (Fig 1A). Each sample before and after the HDAC assay was examined by immunoblotting (Fig 1B). The band intensities of Ac-RAD52 before the HDAC assay were almost the same between the samples incubated in the presence of $H_2O$ and $D_2O$ (Fig 1B and 1C). This result means that, unless water molecules are directly involved in the chemical reaction, the differences in the hydrogen isotopes of water molecules surrounding protein molecules have almost no effect on the chemical reaction rates due to indirect phenomena, such as affecting the movement of the reacting molecules. In contrast, after the HDAC assay, the relative intensity of the Ac-RAD52 band was more efficiently decreased in the presence of $H_2O$ than in the presence of $D_2O$ (Fig 1B and 1C). Therefore, the presence of $D_2O$ decreased the SIRT3-mediated deacetylation reaction of Ac-RAD52.

Next, we conducted the same experiments using $H_2O$ and $D_2O$ obtained from several different manufacturers (S2A and S2B Fig). As a result, there were no significant differences in the deacetylation rates among the three types of $H_2O$-containing buffers or between the two types of $D_2O$-containing buffers. Therefore, regardless of the types of $H_2O$ or $D_2O$, the effect on the HDAC reaction rate was solely caused by the difference in the isotope between $H_2O$ and $D_2O$. When the SIRT3-mediated RAD52 deacetylation assays were performed at 42˚C, the kinetic isotope effects disappeared (Fig 1D–1G). This temperature dependency suggests that the quantum tunneling effect might be involved in the SIRT3-mediated RAD52 deacetylation (Fig 1H).

## Kinetic isotope effects on SIRT3-mediated histone deacetylation

In eukaryotes, including humans, genomic DNA is wrapped around histone proteins to form nucleosomes, which are further folded into chromatin, and thereby the DNA is accommodated within the cell [53]. Histone proteins form an octamer containing two each of the four protein subunits, H2A, H2B, H3, and H4. Histone acetylation modifications are catalyzed by HATs, while HDACs remove these modifications [54]. To examine the kinetic isotope effects on histone deacetylation, we used p300, SIRT3, and unmodified recombinant histone octamer proteins as their substrates [38, 55]. At first, histone octamer proteins are incubated with p300 and $^{14}$C-labeled Acetyl Coenzyme A (Ac-CoA) in reaction buffer containing $H_2O$ or $D_2O$, for histone acetylation at 30˚C (HAT assay). A portion of each reaction mixture containing the acetylated histones is then incubated with SIRT3 and $NAD^+$ in reaction buffer containing $H_2O$ or $D_2O$, respectively (HDAC assay) (Fig 2A). Each sample before and after the HDAC assay is separated by sodium dodecyl sulfate (SDS)-polyacrylamide gel electrophoresis (PAGE), and detected with Coomassie Brilliant Blue (CBB) staining and autoradiography. The total proteins are stained by CBB, and the acetylated proteins are detected by autoradiography. The relative quantities of the mixture of histones H3, H2A, and H2B (top bands), and H4 (bottom bands) are quantified. The kinetic isotope effects were detected at 15˚C (Fig 2B–2E), but disappeared at 42˚C (S3 Fig). Therefore, the temperature dependency of the kinetic isotope effects suggests that the quantum tunneling effect might be involved in the SIRT3-mediated deacetylation of histone proteins.

## Kinetic isotope effect on proteolytic protein cleavage

Protein cleavage by protease family proteins is an enzymatic reaction mediated by hydrolysis. The protease family proteins are involved in diverse biological functions, such as protein degradation, signal transduction, apoptosis, autophagy, immunity, and viral infection [8–12]. We

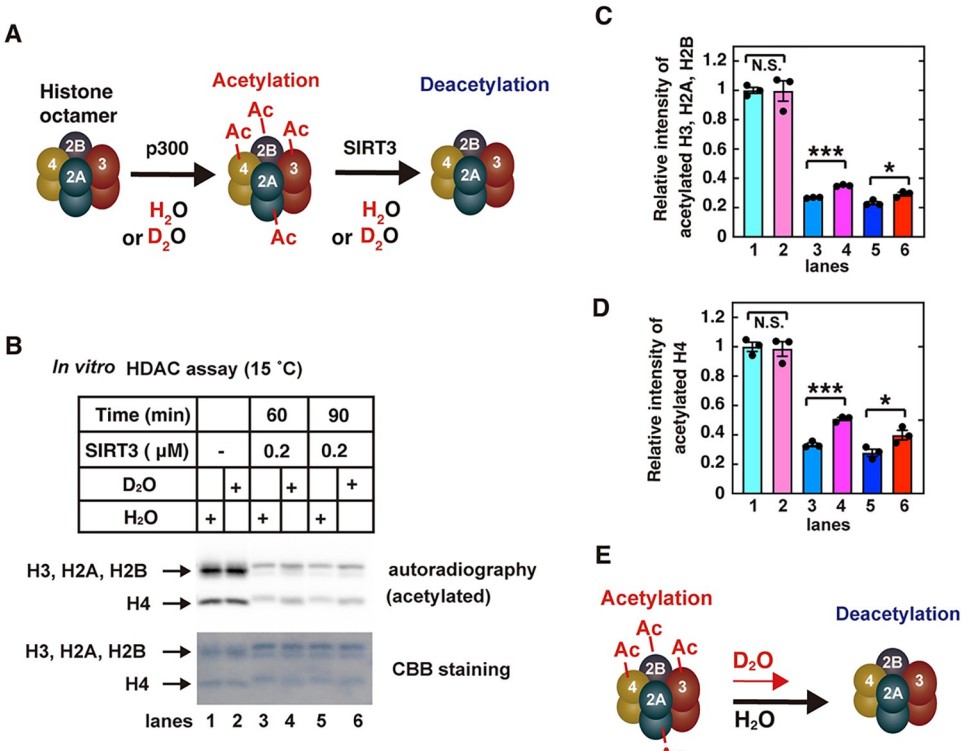

**Fig 2. Kinetic isotope effect on histone protein deacetylation reactions. A)** Schematic representation of the histone deacetylation assay. (**B**) *In vitro* acetylation and deacetylation assays of histone proteins were performed, as described in the Materials and Methods. The deacetylation reactions were performed with the indicated amounts of SIRT3 at 15˚C for the indicated times. The reaction mixtures were subjected to SDS-PAGE, followed by CBB staining (bottom) and autoradiography (top, acetylated proteins). (**C** and **D**) The relative band intensities of acetylated H3, H2A, and H2B (C) and acetylated H4 (D). The graph shows the mean values and standard errors of the mean from 3 independent experiments, with dots of each data values. The samples connected by lines were compared (*$P < 0.05$, ***$P < 0.001$ and N.S., not significant by an unpaired Student´s t-test). (**E**) Schematic representation of the experimental results about the kinetic isotope effect on protein deacetylation reaction.

examined the involvement of the quantum effect on protein cleavage by caspase-3, a protease that functions in apoptosis induction [39] (Fig 3A–3E). The caspase-3 substrate is the DEVD peptide sequence, and its derivative, Ac-DEVD-AMC, generates fluorescence upon cleavage [11, 39] (Fig 3D). The time-course of DEVD cleavage was monitored with $H_2O$ or $D_2O$ in the reaction buffer, at 23˚C or 37˚C (Fig 3A and 3B). At both reaction temperatures, the reaction rate was lower with $D_2O$ than with $H_2O$ (Fig 3A, 3B and 3E). The slope ratio in the presence of $H_2O$ to $D_2O$ was 1.75 at 23˚C, and decreased to 1.48 at 37˚C (Fig 3A and 3B). Therefore, the temperature dependency of the difference in the kinetic isotope effects suggests that the quantum tunneling effect might be involved in the caspase-3-mediated peptide cleavage reaction.

## Kinetic isotope effect on nuclease-mediated DNA cleavage

DNA cleavage by nuclease family proteins is also mediated by hydrolysis (Fig 4A). The nuclease family proteins play a variety of roles, including DNA repair and recombination [3–7]. We examined the involvement of the quantum effect on DNA cleavage by a nuclease, I-SceI [56]. The cleavage rates by I-SceI against a linearized double-stranded DNA containing the specific cleavage sequence were examined in reaction solutions containing $H_2O$ or $D_2O$ (Fig 4B–4D). Kinetic isotope effects were detected at 15˚C, but disappeared at 37˚C (Fig 4B–4E). Thus, the

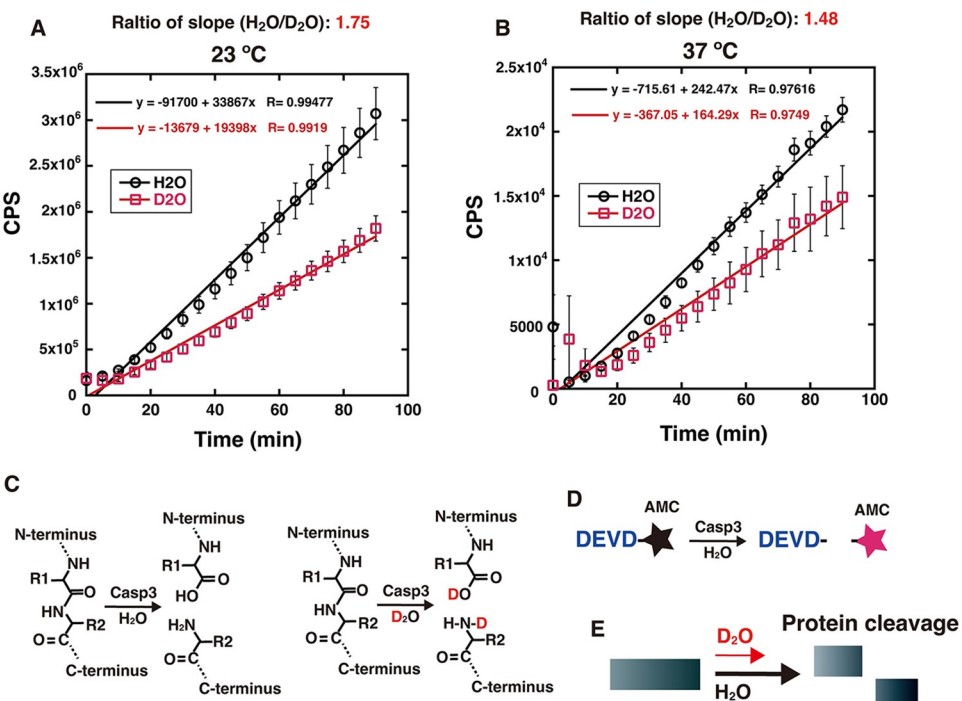

**Fig 3. Kinetic isotope effect and its temperature dependency in peptide-bond cleavage by caspase-3.** (**A** and **B**) *In vitro* caspase-3-mediated peptide-bond cleavage assays were performed at 23°C (A) or 37°C (B) in the presence of $H_2O$ or $D_2O$, as described in the Materials and Methods. The graph shows the mean values and standard errors of the mean from 3 samples, with linear curve fitting. The ratios of the slopes of the linear curve fitting of $H_2O$ to $D_2O$ are shown at 23°C (A) and 37°C (B). (**C**) Schematic representation of peptide bond hydrolysis in the presence of $H_2O$ or $D_2O$. (**D**) Schematic representation of the caspase-3-mediated peptide cleavage assay using the Ac-DEVD-AMC substrate. (**E**) Schematic representation of the experimental results for the kinetic isotope effect on protein cleavage.

kinetic isotope effect and the temperature dependency also suggest that the quantum tunneling effect might be involved in the I-SceI-mediated DNA cleavage reaction.

## Heavy water severely impacts HR repair required for protection of cell survival

To clarify the biological consequences of the quantum effects, we examined the isotope effects in cells with respect to the kinetic isotope effects of the SIRT3 and I-SceI enzymes detected in the *in vitro* experiments. For this purpose, we performed the HR repair assay using human DR-GFP reporter cells (Fig 5A and 5B) [40, 41] The I-SceI sequence is absent in the genome of normal human cells, and the DR-GFP reporter cells contain a reporter DNA cassette containing the I-SceI sequence in the genome. In the HR repair assay, a DSB is specifically induced within the reporter cassette when the I-SceI enzyme is produced by the expression plasmid transfected in the cells, and normal GFP genes are expressed when the DSB site is repaired by HR (Fig 5B). The HR repair frequency is dependent on the DSB induction frequency by I-SceI (Fig 5C). Because SIRT3-mediated RAD52 deacetylation is involved in HR repair, the depletion of RAD52 or SIRT3 decreased the HR repair efficiency (Fig 5D) [37, 48]. In the *in vitro* experiments, the presence of $D_2O$ reduced the reaction rate by at most 1/2. Unexpectedly, we found that HR repair was almost completely inhibited in cells cultured in medium containing $D_2O$ (Fig 5C and 5D). HR repair is also required for normal cell growth, and cell death is induced by strongly inhibiting HR repair factors [57–59]. This might be one of the reasons for

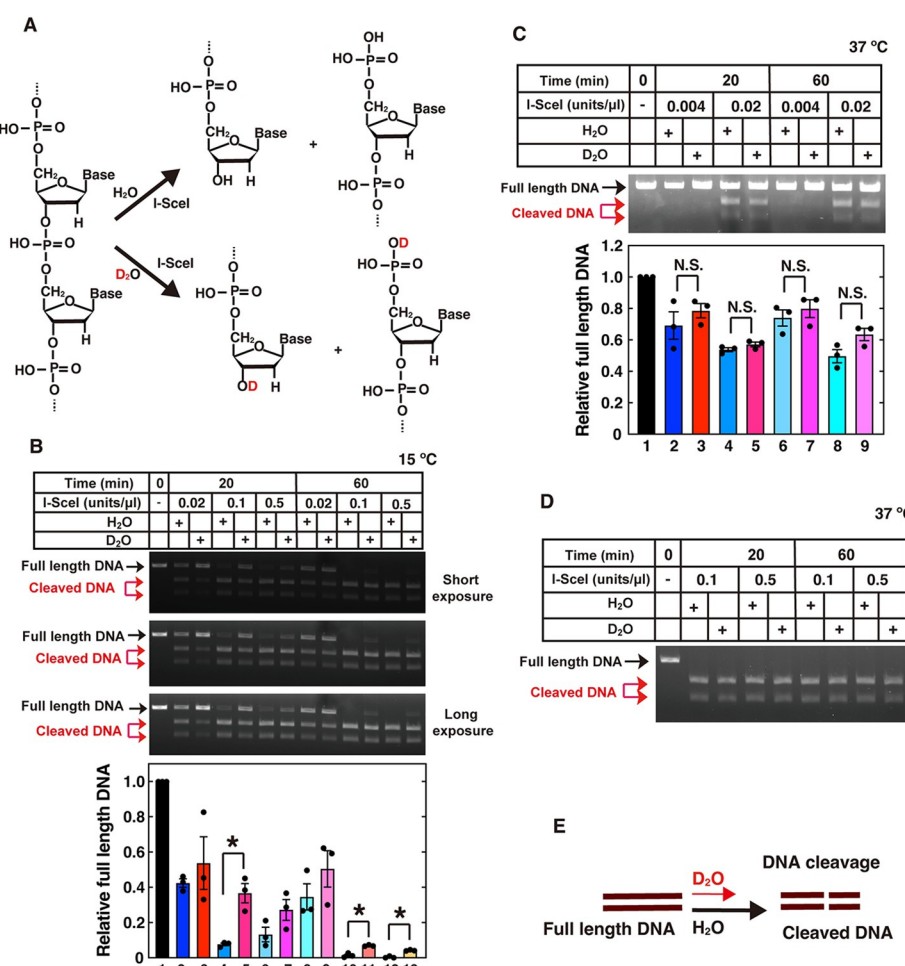

**Fig 4. Kinetic isotope effect and its temperature dependency in DNA cleavage by I-SceI nuclease.** (**A**) Schematic representation of phosphodiester bond hydrolysis by I-SceI in the presence of $H_2O$ or $D_2O$. (**B** to **D**) *In vitro* I-SceI-mediated DNA cleavage assays were performed at 15˚C (B) or 37˚C (C and D) for the indicated times in the presence of $H_2O$ or $D_2O$, as described in the Materials and Methods. (**B** and **D**) The relative band intensities of full-length DNA are shown in the graph. The mean values and standard errors of the mean from 3 independent experiments were plotted, with dots of each data values. The samples connected by lines were compared (*$P$ <0.05 and N.S., not significant by an unpaired Student's t-test). (**E**) Schematic representation of the experimental results for the kinetic isotope effect on DNA cleavage.

our results that cell death was induced by exposure to $D_2O$ in various types of normal and cancer cells, in a $D_2O$ concentration-dependent manner, consistent with previous reports (Fig 6A–6D) [30, 34, 35].

## Heavy water progressively and strongly inhibits the accumulation of DSB repair proteins at the DSB site throughout the DNA repair process

To elucidate the mechanism by which HR repair is inhibited in cells exposed to heavy water, we next examined the effect of heavy water on the accumulation of the DSB repair proteins ATM, 53BP1, BRCA1, and RAD51 at DSB sites after irradiation. ATM is required for HR repair and is also involved in the recruitment of several DSB repair proteins, such as RAD52, at DSB sites [37, 60–64]. 53BP1 also accumulates at DSB sites early after DSB induction, and regulates the repair pathway choice between HR and NHEJ [65–68]. Subsequently, BRCA1

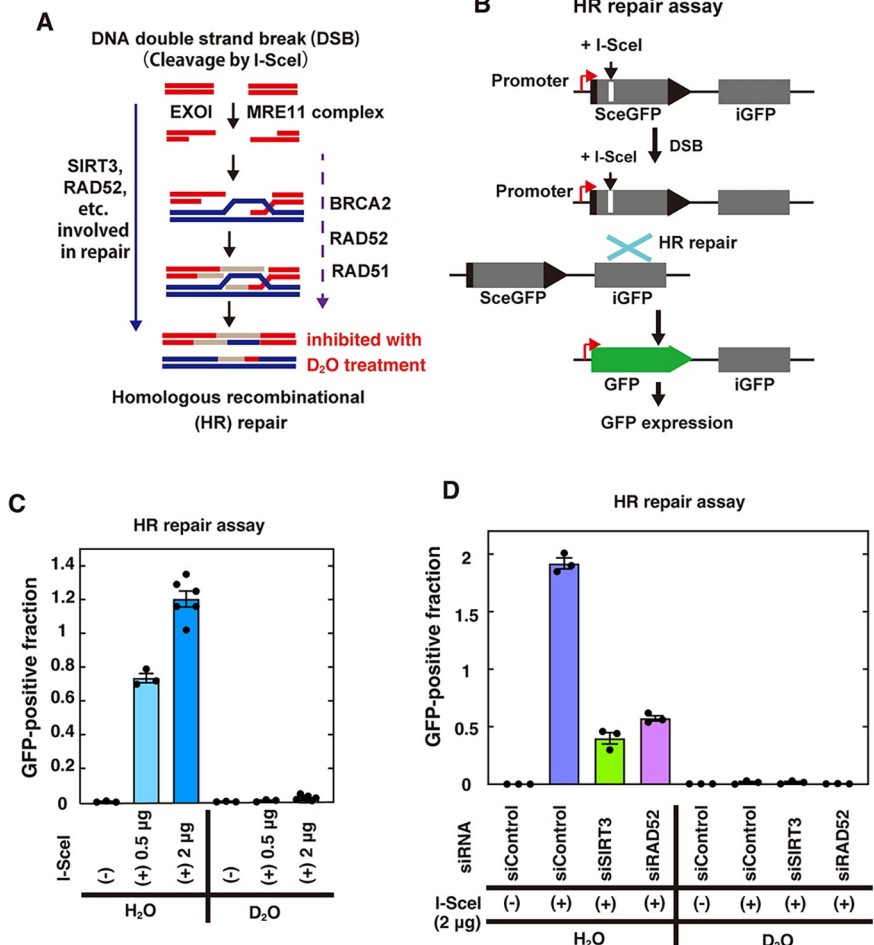

**Fig 5. Isotope effects of D$_2$O on HR repair in human cells.** (**A**) Schematic representation of homologous recombination (HR) repair of an I-SceI-induced DSB site. (**B**) Schematic representation of the I-SceI-induced HR repair reporter assay. HR repair at I-SceI-induced specific DSB sites produces GFP-positive cells. (**C** and **D**) I-SceI-based reporter assays for HR were performed in HeLa pDR-GFP cells cultured in medium containing H$_2$O or D$_2$O and transfected with the indicated amount of the I-SceI plasmid, as described in the Materials and Methods. The graph shows the mean values and standard errors of the mean from 3 or 6 samples, with dots of each data values (2 μg of I-SceI in Fig 6C).

accumulates at DSB sites and participates in the recruitment of RAD51, which is essential for HR repair [69]. BRCA1 promotes homologous recombination reactions mediated by RAD51 [66, 70].

In cells exposed to heavy water, phosphorylated ATM foci [71] were observed at 1 hr after irradiation, but the co-localization with DSB sites was greatly reduced. Because ATM phosphorylation occurred, we assumed that ATM should accumulate at DSB sites immediately after irradiation. Therefore, we examined the effect of heavy water 10 min after irradiation, and found that most of the ATM was still localized at the DSB sites (Fig 7). Similarly, in cells exposed to heavy water, the co-localization of 53BP1 foci with DSB sites was greatly reduced after 1 hour of irradiation. In contrast, 10 minutes after irradiation, the percentage of 53BP1 foci co-localizing with DSB sites was higher in the presence of heavy water than at 1 hour after irradiation (Fig 8). Six hours after irradiation, the accumulation of RAD51 and phosphorylated BRCA1 [72] at DSB sites was almost completely inhibited in cells exposed to heavy water (Fig

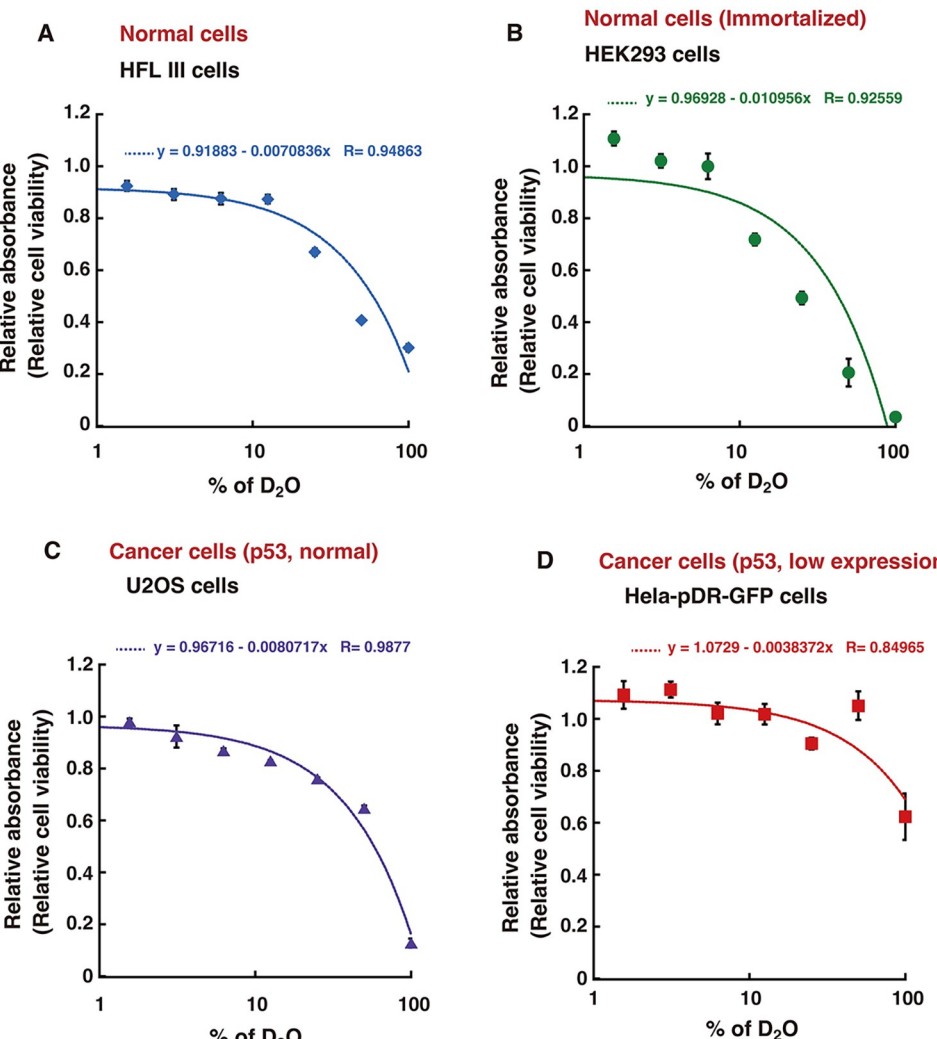

**Fig 6. Cytotoxicity of D₂O in human cells.** (**A** to **D**) HFL III (A), HEK293 (B), U2OS (C), and HeLa pDR-GFP (D) cells were cultured in media containing various concentrations of D₂O for 6 days. Cell survival was examined by an MTT assay, as described in the Materials and Methods. The graph shows the mean values and standard errors of the mean from 3 samples, with linear curve fitting.

9). These results indicate that the accumulation of proteins involved in DNA repair at DSB sites is more strongly inhibited from the upstream to the downstream reaction steps of DSB repair, resulting in the inhibition of HR repair.

## Heavy water inhibits both single-strand annealing (SSA) and non-homologous end joining (NHEJ) repairs in human cells

In addition to HR repair, other DNA repair mechanisms also involve hydrolytic enzymatic reactions, such as DNA cleavage by repair enzymes. For example, DNA end resection and DNA end processing by nucleases are involved in SSA repair, in which homologous repeated sequences that flank the DSB site are annealed (Fig 10A) [4, 73]. In NHEJ, which repairs DNA by directly binding DSB ends without annealing of homologous DNA sequences, nucleases, such as the MRE11 complex and ARTEMIS, are also involved in the repair (Fig 10B) [74–76].

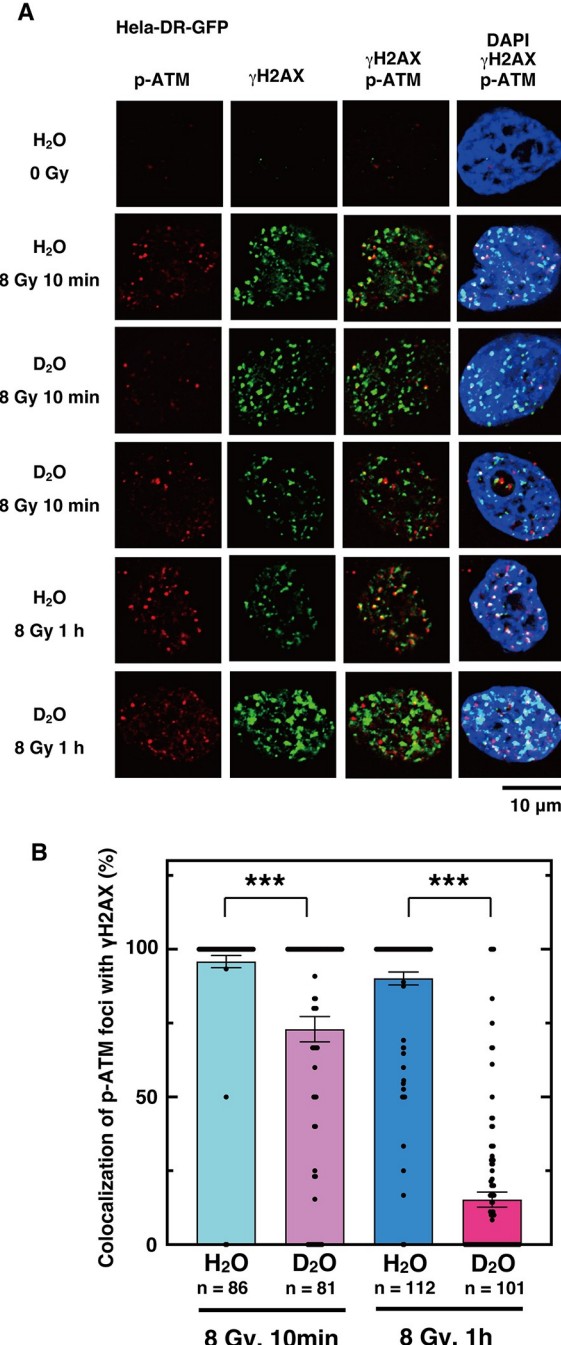

**Fig 7. ATM is activated at the DSB site in the presence of D₂O, but dissociates from the DSB site.** (A) HeLa pDR-GFP cells were cultured in fresh media containing $H_2O$ or $D_2O$ for 1 h before irradiation with γ-rays (8 Gy). Cells without irradiation (0 Gy), or at 10 min or 1 h after irradiation were subjected to immunofluorescent staining with anti-phospho-ATM at Ser1981 (green) antibody, anti-γH2AX (red) antibody, and DAPI (blue). (**B**) The percentages of phospho-ATM foci colocalized with γH2AX were calculated by dividing the number of phospho-ATM foci colocalized with γH2AX by the total number of phospho-ATM foci in each cell. The graph shows the mean of the calculated percentages of the number of cells indicated for each condition, with dots representing each data value. Error bars indicate the standard error of the mean. Asterisks indicate statistically significant differences between the samples connected by lines. (***, $p < 0.001$ by an unpaired Student´s t-test performed with Kaleida Graph 5.0).

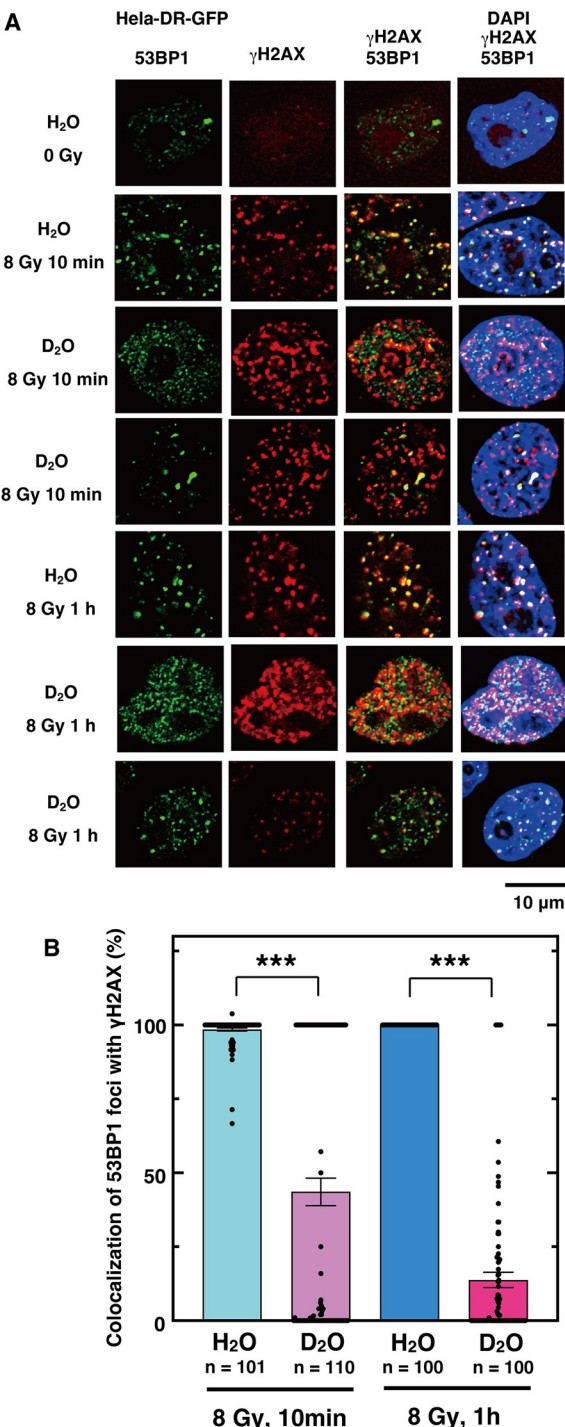

**Fig 8. Accumulation of 53BP1 to DSB sites is partially inhibited by $D_2O$.** (**A**) HeLa pDR-GFP cells were cultured in fresh media containing $H_2O$ or $D_2O$ at 1 h before irradiation with $\gamma$-rays (8 Gy). Cells without irradiation (0 Gy), or at 10 min or 1 h after irradiation were subjected to immunofluorescent staining with anti-53BP1 (green) antibody, anti-$\gamma$H2AX (red) antibody, and DAPI (blue). (**B**) The percentages of 53BP1 foci colocalized with $\gamma$H2AX were calculated by dividing the number of 53BP1 foci colocalized with $\gamma$H2AX by the total number of 53BP1 foci in each cell. The graph shows the mean of the calculated percentages of the number of cells indicated for each condition, with dots representing each data value. Error bars indicate the standard error of the mean. Asterisks indicate statistically significant differences between the samples connected by lines. (***, $p<0.001$ by an unpaired Student´s t-test performed with Kaleida Graph 5.0).

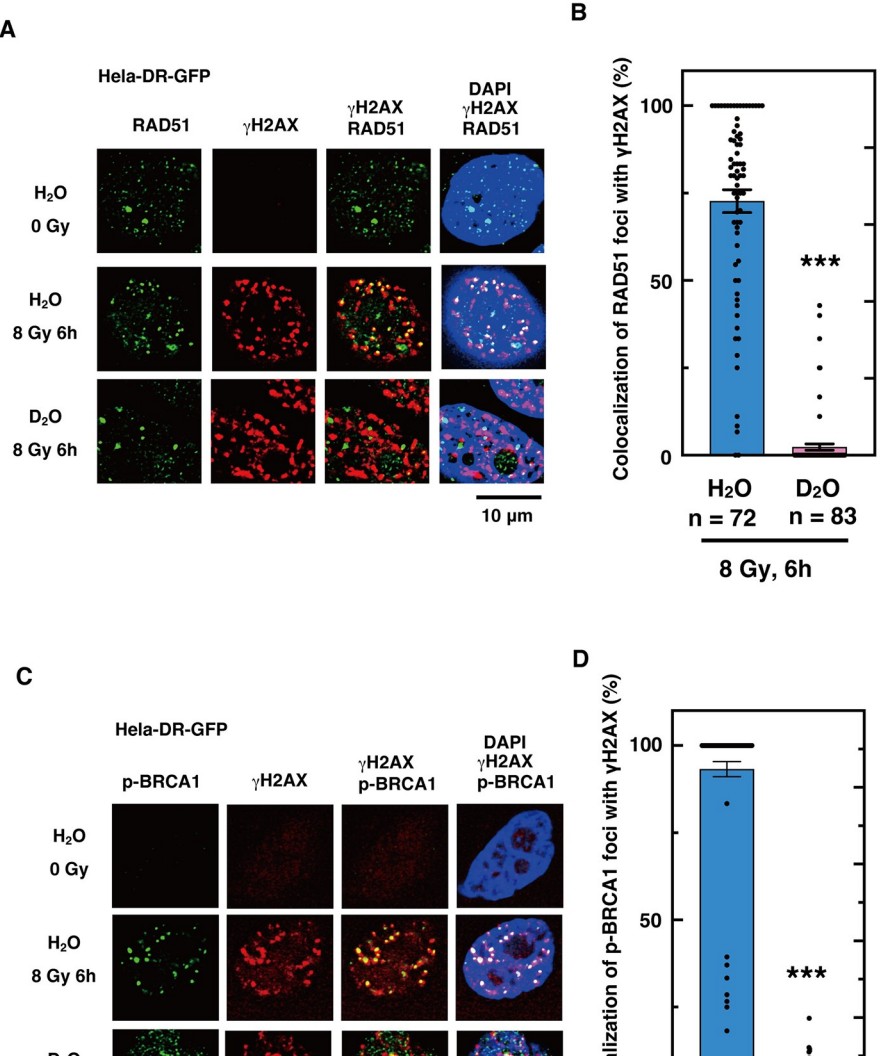

**Fig 9. Accumulation of phosphorylated BRCA1 and RAD51 to DSB sites is severely inhibited by $D_2O$.** (**A** and **C**) HeLa pDR-GFP cells were cultured in fresh media containing $H_2O$ or $D_2O$ immediately before irradiation with γ-rays (8 Gy). Cells without irradiation (0 Gy) or at 6 h after irradiation were subjected to immunofluorescent staining. (**A**) Anti-RAD51 (green) antibody, anti-γH2AX (red) antibody, and DAPI (blue) were used. (**C**) Anti-phospho-BRCA1 at Ser1254 (green) antibody, anti-γH2AX (red) antibody, and DAPI (blue) were used. (**B** and **D**) The percentages of RAD51 (B) or phospho-BRCA1 (D) foci colocalized with γH2AX were calculated by dividing the number of RAD51 (B) or phospho-BRCA1 (D) foci colocalized with γH2AX by the total number of RAD51 (B) or phospho-BRCA1 (D) foci in each cell, respectively. The graph shows the mean of the calculated percentages of the number of cells indicated for each condition, with dots representing each data value. Error bars indicate the standard error of the mean. Asterisks indicate statistically significant differences between the samples containing $H_2O$ or $D_2O$. (***, $p<0.001$ by an unpaired Student´s t-test performed with Kaleida Graph 5.0).

Therefore, we also examined the effects of exposing cells to heavy water on SSA and NHEJ repairs. Similar to the reporter assay for HR repair, the repair of DSB sites induced by I-SceI was analyzed using reporter assay systems for SSA or NHEJ (Fig 10C and 10D). The results

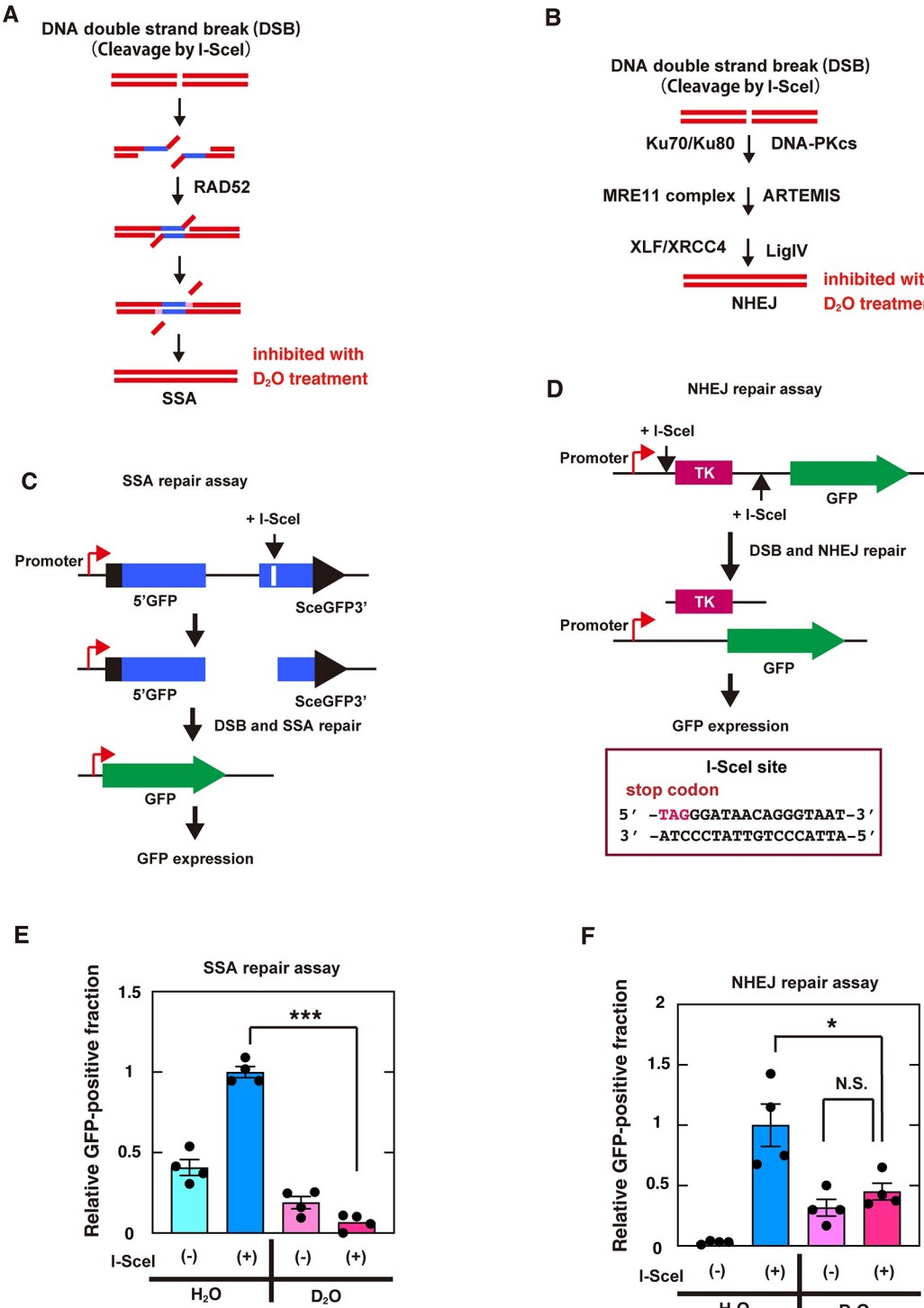

**Fig 10. Isotope effects of D₂O on SSA and NHEJ repairs in human cells.** (**A** and **B**) Schematic representations of SSA (A) and NHEJ (B) repairs of an I-SceI-induced DSB site. (**C** and **D**) Schematic representations of the I-SceI-induced SSA (C) and NHEJ (D) repair reporter assays. SSA or NHEJ repair at I-SceI-induced specific DSB sites produces GFP-positive cells. (**E** and **F**) I-SceI-based reporter assays for SSA (E) or NHEJ (F) were performed in HEK293 SAGFP or H1299 dA3-1#1 cells, respectively, cultured in medium containing H₂O or D₂O with (+) or without (-) transfection of the I-SceI plasmid, as described in the Materials and Methods. The graph shows the mean values and standard errors of the mean from 4 samples, with dots representing each data value. The samples connected by lines were compared (*$P < 0.05$, ***$P < 0.001$ and N.S., not significant by an unpaired Student´s t-test).

showed that both SSA and NHEJ repairs were significantly inhibited in cells exposed to heavy water (Fig 10E and 10F).

## Isotope effect by $D_2O$ on histone acetylation and transcription in cells

Histone acetylation in cells is altered by the balance between acetylase and deacetylase activities, which are involved in transcriptional regulation (Fig 11A) [77–80]. Therefore, when histone deacetylases are inhibited, histone acetylation is induced, and thereby transcription is increased (Fig 11A). Consistent with the kinetic isotope effect on the *in vitro* histone deacetylation, the induction of histone acetylation was detected when cells were exposed to $D_2O$ (Fig 11B). A global transcriptional analysis with RNA-seq revealed that overall transcription changed much more by a $D_2O$-treatment than by an siRNA-treatment against SIRT3 (Fig 11C, S4 and S5 Figs). On average, the $D_2O$-treatment increased transcription far more than the

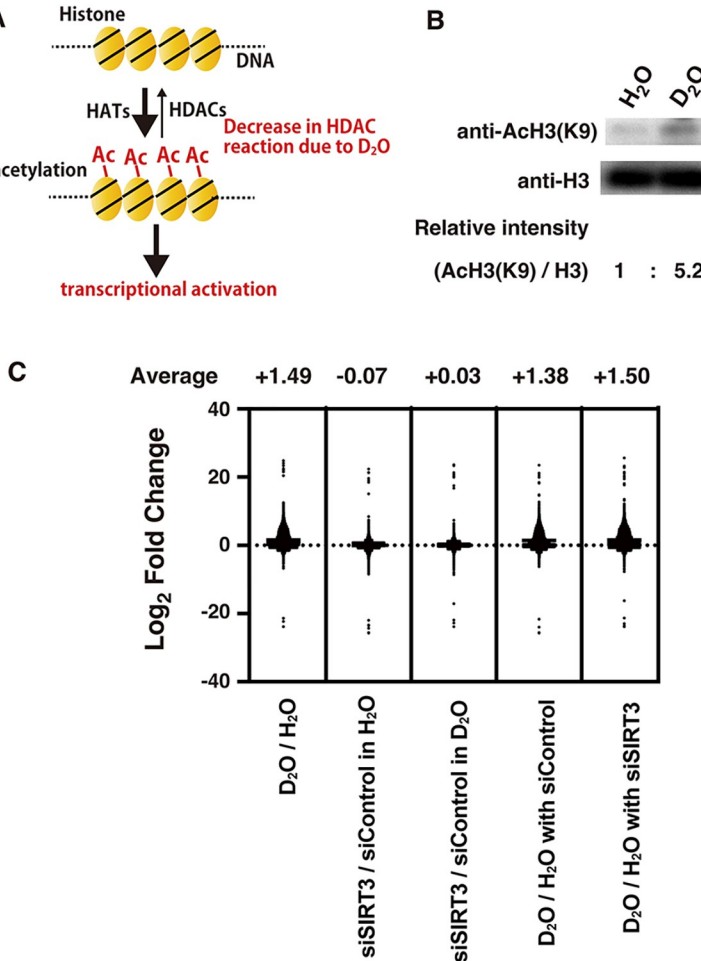

**Fig 11. Isotope effects of $D_2O$ on histone acetylation and transcription in human cells.** (**A**) Schematic representation of the regulation of histone acetylation-mediated transcriptional activation by HATs and HDACs and the isotope effect by $D_2O$. (**B**) U937 human cells were cultured in medium containing $H_2O$ or $D_2O$ for 2 h. The cell extracts were subjected to immunoblotting analyses with the indicated antibodies. The relative band intensities of acetylated H3 at K9 (AcH3(K9)) bands normalized to those of the H3 bands are shown below the immunoblots. (**C**) RNA-seq analyses were performed with HeLa pDR-GFP human cells cultured in medium containing $H_2O$ or $D_2O$ for 5 h, as described in the Materials and Methods. The dots in the graph show the mean values of $Log_2$ fold change of each gene from 6 samples. The average values of $Log_2$ fold change of whole genes are shown in each of the comparison conditions.

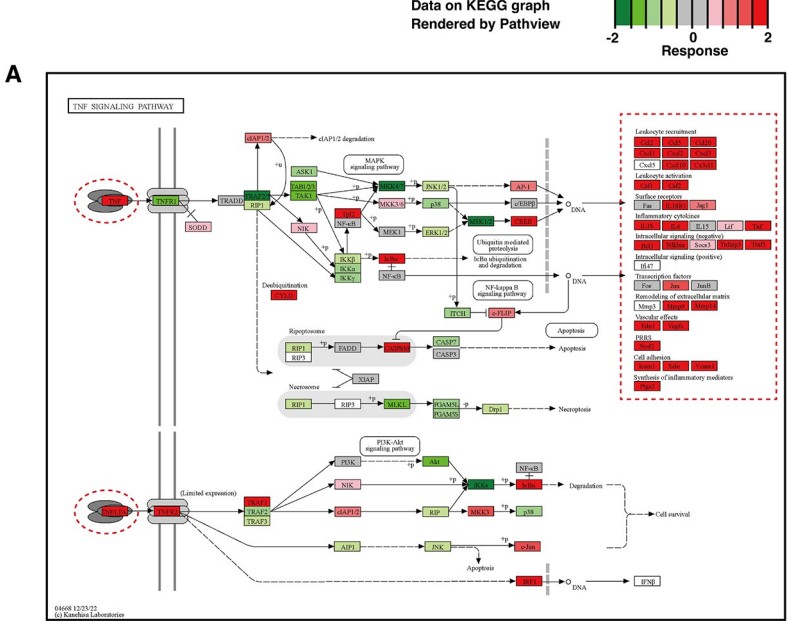

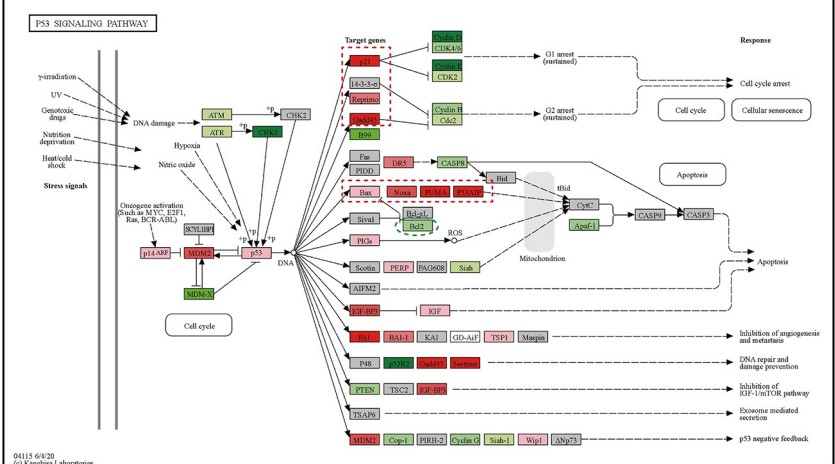

**Fig 12. Effects of D₂O on expression levels of cellular genes involved in "TNF signaling pathway" and "p53 signaling pathway".** (**A** and **B**) The RNA-seq data shown in S4 Fig were used. Expression levels of each gene were visualized on KEGG pathway maps of "TNF signaling pathway" (**A**) and "p53 signaling pathway" (**B**), as described in the Materials and Methods. The red and green colors, according to shading, show an increase and decrease in gene expression, respectively, with D₂O treatment compared to H₂O treatment. (**A**) Characteristic groups of genes with increased or decreased expression are surrounded by red or green dashed lines, respectively.

siRNA-treatment against SIRT3 in cells. The RNA-seq data analysis revealed that there were similarities in the groups of genes with transcription altered by D₂O-treatment under both the mock and siRNA-treatment conditions (S6–S8 Figs). The gene expressions from the RNA-seq data were analyzed [45, 47] and are shown in color on KEGG pathway maps [46] (Fig 12A and 12B, S9–S18 Figs). The D₂O treatment induced the expression of genes involved in cellular defenses and cytokines (Fig 12A, S9 and S10 Figs), but decreased the transcription of genes involved in DNA transactions and the cell cycle (S11–S18 Figs). Among the genes involved in homologous recombination (HR), the expression of most of the genes required for HR repair,

such as MRN complexes [3, 4], was decreased by $D_2O$-treatment, whereas the expression of SYCP3 [81], which suppresses HR repair, was increased (S14 Fig). Exposure to heavy water also reduced the expression of most genes involved in other DNA repair pathways, including nucleotide excision repair (NER) [82–84], base excision repair (BER) [85, 86], mismatch repair (MMR) [87], and the Fanconi anemia (FA) DNA repair pathway [88, 89] (S15–S18 Figs). The expression of genes regulated by p53, such as p21 and GADD45 [90], which suppress cell cycle progression, and PUMA and Noxa [90], which induce apoptosis, a mechanism of programmed cell death, was increased by $D_2O$-treatment (Fig 12B, S10–S13 Figs). Conversely, the expression of Bcl2 [90], which suppresses apoptosis, was decreased by $D_2O$-treatment (Fig 12B). These effects of $D_2O$ on cellular transcription were similar to those of chemical HDAC inhibitors, which induce the expression of p21, GADD45, and pro-apoptotic genes and reduce the expression of pro-survival and cell cycle genes [54, 91].

### Isotope effect by $D_2O$ on p53-dependent apoptosis and necrosis

The RNA-seq data revealed that $D_2O$-treatment induced the expression of genes related to apoptosis induction in human cells. Consistently, apoptosis was actually induced by $D_2O$-treatment in human cells (Fig 13A). Another cell death pathway, necrosis, was also activated by $D_2O$-treatment (Fig 13B). In addition, since both apoptosis and necrosis were decreased by about half in a p53-null background (Fig 13C and 13D), both cell death pathways induced by $D_2O$ were partially dependent on p53. This p53 dependency is consistent with the RNA-seq analysis results.

### Temperature-dependence of isotope effect of heavy water in human cells

Similar to the reasons for the kinetic isotope effect on the *in vitro* enzymatic reactions, the isotope effect of heavy water on cells can be attributed to two possible reasons: the difference in the vibrational potential energies and the difference in the quantum tunneling effects of hydrogen isotopes. If there is a temperature dependence of the isotope effect caused by heavy water in cells, similar to the temperature dependence of the kinetic isotope effect *in vitro*, then the quantum tunneling effect might be involved in the isotope effect in cells. Therefore, we investigated whether temperature affects the isotope effect caused by heavy water on cell proliferation, in the temperature range at which human cells can grow.

When human cells were cultured in medium with a 20% or 50% concentration of heavy water, elongated cells with abnormal morphology were observed. With the 20% concentration of heavy water, the morphology was almost normal when cells were cultured at 39°C, but abnormal morphology was observed as the temperature decreased (Fig 14A). In addition, the presence of heavy water tended to suppress cell proliferation as the temperature decreased (Fig 14B). Thus, the isotope effect by heavy water on human cells showed a temperature dependence similar to the *in vitro* effects.

## Discussion

### Isotope effects by heavy water on hydrolytic enzyme reactions and cellular functions

In this study, we have presented results of the kinetic isotope effects by heavy water on hydrolytic enzyme chemical reactions: "deacetylation reaction", "DNA cleavage reaction", and "protein cleavage reaction", which are important for life phenomena. In our *in vitro* experiments, the rate reduction by $D_2O$ for the RAD52 deacetylation and I-SceI cleavage reactions was at most 1/2, but the experimental results of HR repair involving these enzyme reactions in cells

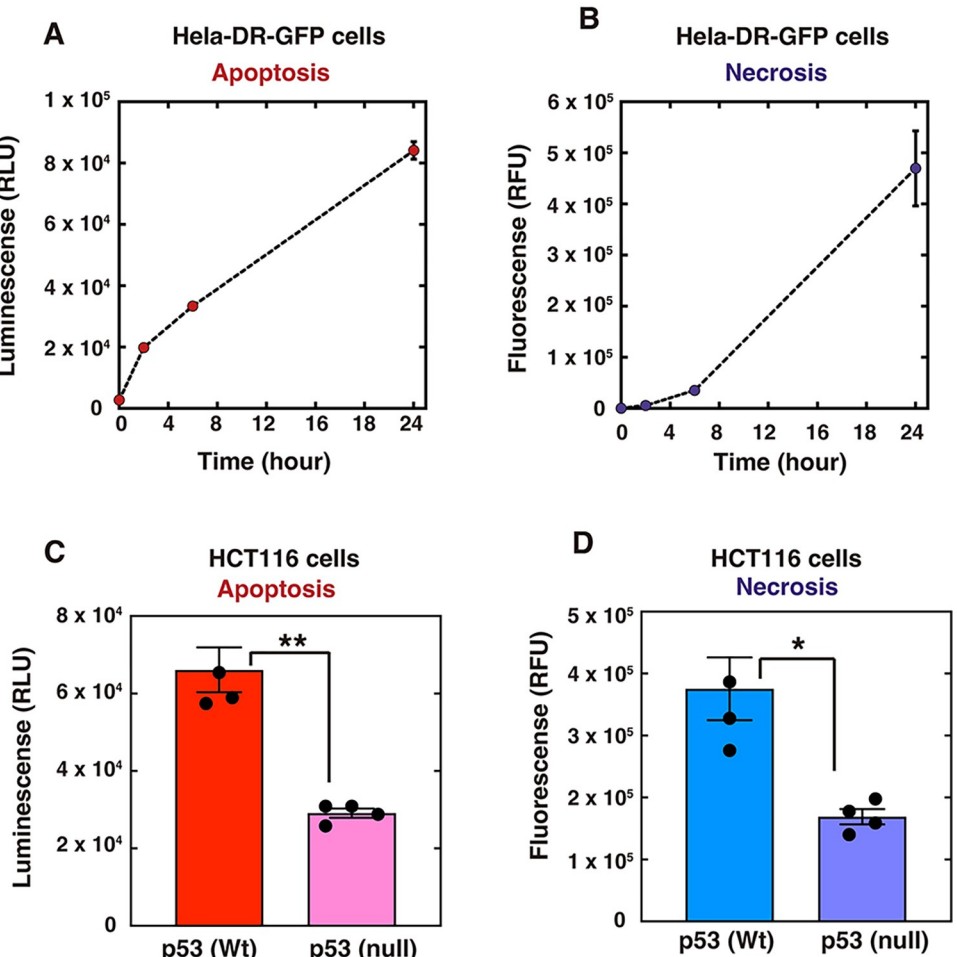

**Fig 13. Apoptosis and necrosis inductions by $D_2O$.** (**A** and **B**) HeLa pDR-GFP cells were cultured in medium containing 100% $D_2O$ for the indicated times. (**C** and **D**) HCT116 (p53 wild type or null) cells were cultured in medium containing 100% $D_2O$ for 24 h. (**A** to **D**) Apoptosis and necrosis assays were performed, as described in the Materials and Methods. The graphs show the mean values and standard errors of the mean from 3 (A and B) or 4 (C and D) samples. (**C** and **D**) The each data values were shown with dots. The samples connected by lines were compared (*$P$ <0.05 and **$P$ <0.01 by an unpaired Student´s t-test).

showed that the recombination frequency was almost completely inhibited by the $D_2O$-treatment. Therefore, kinetic isotope effects of $D_2O$ on cells were much greater than those expected from the *in vitro* experiments. This could be attributed to the fact that cellular reactions are successive reactions by various enzymes, so the final effect might appear as a synergistic reduction in the rates of individual enzymatic reactions due to the kinetic isotope effects. For example, in HR repair reactions, nucleases such as the MRE11 complex and ExoI, which cleave the bonds of DSB-terminal bases [3, 4], are expected to be affected by the kinetic isotope effects in a similar manner to the I-SceI nuclease. In addition to RAD52 deacetylation, various other deacetylation reactions are involved in HR repair, and are also expected to be influenced by the kinetic isotope effects. Multiple ATPases, such as RAD51, also perform HR reactions using energy produced by ATP hydrolysis [3]. Although the kinetic isotope effect in the ATPase-mediated ATP hydrolysis reaction has not been experimentally demonstrated in this study, this reaction may also be affected. Thus, there are many reaction steps in HR repair that could be affected by the kinetic isotope effects. If there are 10 reaction steps subjected to the

**A**

**Concentration of D₂O**

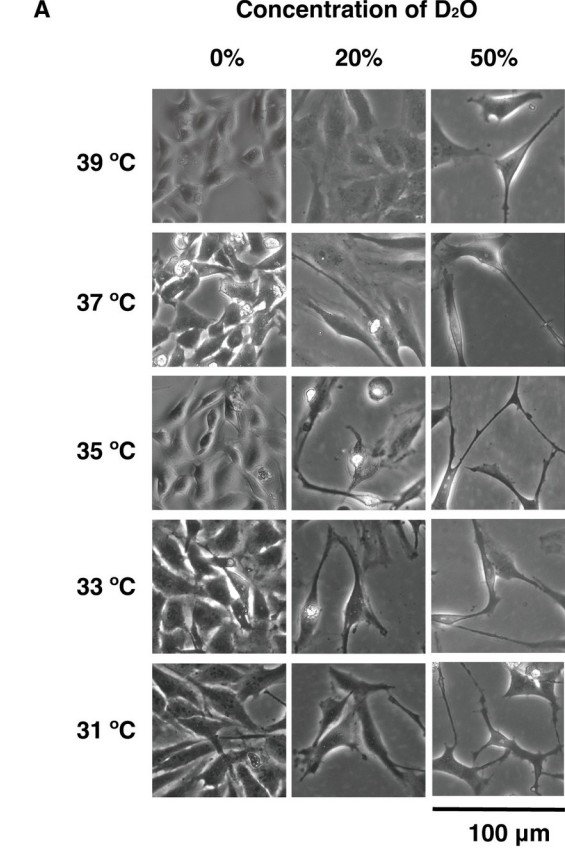

**B**

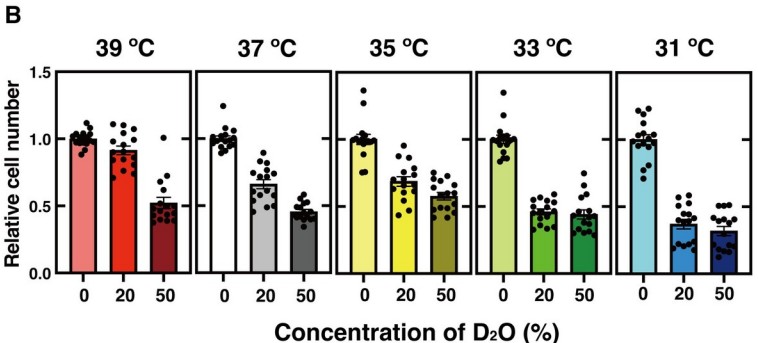

**Fig 14. Temperature-dependent effects of D₂O on cell proliferation.** (**A**) HeLa pDR-GFP cells cultured at 37°C to 100% confluency in a 10 cm plastic dish were stripped with 1 ml of trypsin solution (0.5 g/L) and suspended in 4 ml of normal cell culture medium. A 0.2 ml portion of this cell suspension was suspended in 2 ml of cell culture medium containing 0, 20, or 50% D₂O, and the cells were incubated in a 3.5 cm plastic dish for 1 week at the indicated temperature. The phase-contrast images of the cells were obtained, as described in the Materials and Methods. (**B**) The cell growth assays were performed, as described in the Materials and Methods. At each temperature, the amounts of cells at 20 and 50% D₂O were calculated relative to the number of cells at 0% D₂O. The graph shows the mean values and the standard error, with dots representing each data value (n = 16).

synergistic isotope effects that reduce the reaction rate by 1/2, then the reaction rate is almost completely impeded by $(1/2)^{10} = 1/1{,}024 = 0.0009765625$. For this reason, it can be inferred that the HR repair reaction in the cell was much more strongly inhibited than the extent expected from the *in vitro* kinetic isotope effect. Our findings that the accumulation of DSB repair-related proteins at DSB sites was more strongly inhibited by the isotope effect of heavy

water from the upstream to downstream steps of DSB repair strongly support the above hypothesis regarding the synergistic nature of kinetic isotope effects.

In addition to HR repair, heavy water inhibited the SSA and NHEJ DSB repair pathways (Fig 10). Hydrolytic enzyme chemical reactions are also involved in other DNA repair mechanisms, such as the NER [82–84], BER [85, 86], MMR [87], and FA DNA repair [88, 89] pathways, suggesting that heavy water might impede all DNA repair pathways. Thus, the ability of heavy water to inhibit multiple DNA repair pathways is a unique feature that distinguishes it from previous reagents that affect only specific DNA repair enzymes [92]. DNA repair is required for cell survival. Therefore, living organisms are equipped with multiple DNA repair mechanisms to protect their survival by compensating for the inhibition or genetic deficiency of a specific DNA repair mechanism by the other DNA repair mechanisms. However, heavy water strongly inhibits multiple repair pathways simultaneously, which may be one of the reasons why it induces cell death.

Histone deacetylation by SIRT3 exhibited a kinetic isotope effect, in which the reaction rate was slower with $D_2O$ than with $H_2O$ in the reaction solution. In relation to this *in vitro* kinetic isotope effect, an *in vivo* isotope effect was observed in which the exposure of human cells to $D_2O$ induced cellular histone acetylation. The addition of histone deacetylase inhibitors to cells induces histone acetylation because the intracellular acetylation equilibrium shifts in the direction of acetylation induction by HATs. The effect of $D_2O$ on intracellular histone acetylation is also presumably due to a change in the histone acetylation balance, by decreasing the deacetylation reaction rate. In relation to the fact that the induction of histone acetylation activates transcription, the addition of $D_2O$ resulted in an overall activation of intracellular transcription. Heavy water is thought to disrupt the acetylation equilibrium of histones due to kinetic isotope effects on the hydrolytic enzyme. Meanwhile, various equilibrium states, such as protein phosphorylation/dephosphorylation [13–15] and microtubule polymerization/depolymerization [93, 94], are regulated in biological phenomena. It is speculated that, by the same mechanism as the disruption of acetylation equilibrium by heavy water, the disruption of various equilibria important for life phenomena by heavy water may generate significant intracellular stress, thereby inducing cell death.

## Quantum-level effects on life phenomena

Quantum biology is an exploratory science that seeks to understand biological life phenomena from the standpoint of quantum mechanics theory. The concept is described in the excellent book " Life on the Edge: The Coming of Age of Quantum Biology " by Jim Al-Khalili and Johnjoe McFadden [27]. For understanding life phenomena at the quantum level, several books and reviews have described imaginative future possibilities regarding applications of quantum biology to various life phenomena [27, 95]. The authors suggested that chemical reactions by biomolecules, which make up life, are ultimately quantum level reactions and might therefore be subject to quantum effects. However, some skepticism remains as to whether such microscopic-level effects by quantum mechanics actually have any meaningful impact on macroscopic life systems in living cells or organisms.

In this study, the *in vitro* kinetic isotope effects caused by hydrogen isotopes could be due to quantum-level changes in both the vibrational potential energy and quantum tunneling probability. The *in vitro* kinetic isotope effects became smaller with increasing reaction temperature, suggesting the involvement of quantum tunneling in proton transfer by hydrolytic enzymes in the lower temperature range. Although the existence of quantum tunneling effects in some enzymes had been shown by *in vitro* experiments [19–24], the biological outcomes attributable to these effects have remained elusive. We have experimentally shown the relevant kinetic isotope effects on cells attributed to $D_2O$.

D$_2$O is reportedly cytotoxic [28–35], but it was unclear whether this was due to phenomena related to quantum effects, since no experimental reports have shown their relevance. Meanwhile, protein labeling using carbon or nitrogen isotopes, which are less prone to quantum tunneling due to their large atomic weights compared to hydrogen, is widely used for proteome analysis by the SILAC method because they have no effects on cells [96]. The substitution of protein atoms with their isotopes is not considered to significantly affect enzyme activities, because the outer electrons important for enzymatic chemical reactions are the same within the isotope. Therefore, the major effect of the hydrogen isotope on cells is thought to be the kinetic isotope effects caused by the quantum mechanisms. In this study, as in the *in vitro* kinetic isotope effects, the hydrogen isotope-induced cytotoxicity also showed temperature-dependence. However, unlike the *in vitro* experiments, the temperature changes may cause heat shock or cold shock responses, and the isotope effects on cells might also be affected by these stress responses in cell-based experiments. Thus, if these side effects occur due to stress responses, then the experimental results in cells cannot simply be interpreted in the same way as the *in vitro* experimental results. Even with the possibility of such side effects, the findings that the isotope effect tends to be stronger at lower temperatures in cells suggest that quantum tunneling effects could also be involved in the biological effects exerted by hydrogen isotopes.

On Earth, ~99.972% of the isotopic abundance of hydrogen is H with an atomic mass of 1, and only ~0.028% of the isotopic abundance is D with an atomic mass of 2 [97]. Thus, in the "origin of life," organisms optimized for H were more likely to prevail, since H constitutes most of the hydrogen on Earth. Therefore, in an environment where only H is present as hydrogen, the existence of quantum effects would probably not have an overtly negative effect on the survival of cells. However, in a world where only D exists as hydrogen, organisms, including humans, would not be able to survive due to its unexpectedly extensive quantum effects.

While the biological effects of high concentrations of heavy water are strong, there are also notable effects when the deuterium content in naturally occurring water is depleted [98–104]. Compared to normal water, deuterium-depleted water (DDW) reduces the growth rate of mammalian cells [98–100]. The inhibitory effect of DDW on cancer cell proliferation has attracted keen attention, from the perspective of cancer treatment [99–102]. DDW also alters gene expression [101, 104] and the amounts of intracellular metabolic products [102, 103]. Thus, like heavy water at high concentrations, DDW affects the same biological phenomena such as cell proliferation and transcription. Therefore, our study of quantum effects may provide a clear explanation for the observed biological effects of DDW, which could also result from quantum-level isotope effects on the numerous hydrolytic enzyme reactions within cells. Based on our findings and those of previous studies, it can be inferred that quantum-level mechanisms, including quantum tunneling, in hydrolytic enzyme reactions must have important effects on the fundamental life phenomena of organisms living on Earth.

## Supporting information

**S1 Text. Supporting information S1 Text.**
(PDF)

**S2 Text. Supporting information S2 Text.**
(PDF)

**S1 Fig. Schematic representations of the concept of this research.** (**A**) Conceptual image of quantum tunneling, in which the chemical reaction of the reactant (brown circle) at the ground state can proceed without reaching the transition state by penetrating through the

energy barrier. (**B**) Conceptual diagram of quantum tunneling with a 1D box barrier model (*1*). The height of the potential barrier (V), the thickness of the potential barrier (L), and the incident wave, reflected wave, and transmitted wave are shown. (**C**) Difference in zero-point vibrational energies between hydrogen (H) and deuterium (D) to explain the kinetic isotope effect (*2–4*). (**D**) Chemical reactions of SIRT3-mediated deacetylation of an acetylated lysine residue in the presence of $H_2O$ or $D_2O$.
(PDF)

**S2 Fig. Kinetic isotope effect of SIRT3-mediated deacetylation of RAD52 in the presence of $H_2O$ and $D_2O$ from different sources.** (**A**) *In vitro* acetylation assays were performed as described in Fig 1, in the presence of $H_2O$ or $D_2O$ purchased from different sources. $H_2O$ #1, $H_2O$ #2, $H_2O$ #3, $D_2O$ #1, and $D_2O$ #2 were used, as described in the Materials and Methods. After the addition of a poly dT 68 mer, an aliquot of the reaction mixture containing the RAD52 protein (final concentration 0.08 μM) was incubated with the indicated amount of SIRT3 in HDAC buffer, prepared with $H_2O$ or $D_2O$ from different manufacturers, at 30˚C for 60 min. The reaction mixtures were subjected to SDS-PAGE, followed by immunoblotting with an anti-acetylated lysine antibody (Ac-RAD52) and an anti-RAD52 antibody (bottom, RAD52). The long and short exposures are also presented for the immunoblotting with an anti-acetylated lysine antibody. (**B**) The relative band intensities of acetylated RAD52 normalized to those of the RAD52 bands are shown in the graph. The mean values and standard errors of the mean from 3 independent experiments were plotted, with dots of each data values. For each SIRT3 concentration, the samples connected by lines were compared (*$P < 0.05$, **$P < 0.01$, **$P < 0.001$ and N.S., not significant by one-way ANOVA with Dunnet's post hoc test with the KaleidaGraph software).
(PDF)

**S3 Fig. Temperature dependency of the kinetic isotope effect on SIRT3-mediated histone deacetylation.** (**A** and **D**) *In vitro* acetylation and deacetylation assays of histone proteins were performed, as described in the Materials and Methods. The deacetylation reactions were performed with the indicated amount of SIRT3 at 42˚C for the indicated times. The reaction mixtures were subjected to SDS-PAGE, followed by CBB staining (bottom) and autoradiography (top, acetylated proteins). (**B**, **C**, **E**, and **F**) The relative band intensities of acetylated H3, H2A, and H2B (B and E) and acetylated H4 (C and F) are shown in the graph. Mean values and standard errors of the mean from 3 independent experiments were plotted, with dots of each data values. The samples connected by lines were compared (N.S., not significant by an unpaired Student's t-test).
(PDF)

**S4 Fig. Heatmap analysis with hierarchical clustering of the most variable 1,000 genes.** Hela pDR-GFP cells untreated (mock) or transfected with siRNA (siControl or siSIRT3) were cultured in medium made with $H_2O$ or $D_2O$ for 5h, and their RNA samples were subjected to an RNA-seq analysis. Six samples were used for each experimental condition. The heatmap was generated with iDEP96, as described in the Materials and Methods.
(PDF)

**S5 Fig. Volcano plots illustrating isotope effects by $D_2O$ on gene expression levels.** The RNA-seq data shown in S4 Fig were used for a volcano plot analysis with the DEG2 function of iDEP96. The $\log_2$ (fold change) versus -log10 (false discovery rate (FDR)) is shown in each plot.
(PDF)

**S6 Fig. Enrichment analysis of gene expression in Hela pDR-GFP cells untreated with siRNA and cultured in the presence of H$_2$O or D$_2$O.** (**A** to **D**) The RNA-seq data shown in S4 Fig were used. The RNA-seq data were analyzed with the DEG2 function of iDEP96. Upregulated and downregulated genes are colored red and green, respectively. (**A**) Heatmap analysis of gene expression differences. (**B** to **D**) Enrichment trees. Enrichment pathway analyses were performed for three categories: GO biological process (**B**), GO cellular component (**C**), and GO molecular function (**D**). The false discovery rate (FDR) is shown and is also represented by the size of the circle.
(PDF)

**S7 Fig. Enrichment analysis of gene expression in Hela pDR-GFP cells treated with control siRNA (siControl) and cultured in the presence of H$_2$O or D$_2$O.** (**A** to **D**) The RNA-seq data shown in S4 Fig were used. The RNA-seq data were analyzed with the DEG2 function of iDEP96. Upregulated and downregulated genes are colored red and green, respectively. (**A**) Heatmap analysis of gene expression differences. (**B** to **D**) Enrichment trees. Enrichment pathway analyses were performed for three categories: GO biological process (**B**), GO cellular component (**C**), and GO molecular function (**D**). The false discovery rate (FDR) is shown and is also represented by the size of the circle.
(PDF)

**S8 Fig. Enrichment analysis of gene expression in Hela pDR-GFP cells treated with siRNA against SIRT3 (siSIRT3) and cultured in the presence of H$_2$O or D$_2$O.** (**A** to **D**) The RNA-seq data shown in S4 Fig were used. The RNA-seq data were analyzed with the DEG2 function of iDEP96. Upregulated and downregulated genes are colored red and green, respectively. (**A**) Heatmap analysis of gene expression differences. (**B** to **D**) Enrichment trees. Enrichment pathway analyses were performed for three categories: GO biological process (**B**), GO cellular component (**C**), and GO molecular function (**D**). The false discovery rate (FDR) is shown and is also represented by the size of the circle.
(PDF)

**S9 Fig. Effects of D$_2$O on expression levels of cellular genes involved in "cytokine-cytokine receptor interaction".** The RNA-seq data shown in S4 Fig were used. Expression levels of each gene were visualized on KEGG pathway map of "cytokine-cytokine receptor interaction", as described in the Materials and Methods. The red and green colors, according to shading, show an increase and decrease in gene expression, respectively, with D$_2$O treatment compared to H$_2$O treatment.
(PDF)

**S10 Fig. Effects of D$_2$O on expression levels of cellular genes involved in "cellular senescence".** The RNA-seq data shown in S4 Fig were used. Expression levels of each gene were visualized on KEGG pathway map of "cellular senescence", as described in the Materials and Methods. The red and green colors, according to shading, show an increase and decrease in gene expression, respectively, with D$_2$O treatment compared to H$_2$O treatment. Characteristic groups of genes with increased expression are surrounded by red dashed lines.
(PDF)

**S11 Fig. Effects of D$_2$O on expression levels of cellular genes involved in "basal transcription factors".** The RNA-seq data shown in S4 Fig were used. Expression levels of each gene were visualized on KEGG pathway map of "basal transcription factors", as described in the Materials and Methods. The red and green colors, according to shading, show an increase and

decrease in gene expression, respectively, with $D_2O$ treatment compared to $H_2O$ treatment.
(PDF)

**S12 Fig. Effects of $D_2O$ on expression levels of cellular genes involved in "DNA replication".** The RNA-seq data shown in S4 Fig were used. Expression levels of each gene were visualized on KEGG pathway map of "DNA replication", as described in the Materials and Methods. The red and green colors, according to shading, show an increase and decrease in gene expression, respectively, with $D_2O$ treatment compared to $H_2O$ treatment.
(PDF)

**S13 Fig. Effects of $D_2O$ on expression levels of cellular genes involved in "cell cycle".** The RNA-seq data shown in S4 Fig were used. Expression levels of each gene were visualized on KEGG pathway map of "cell cycle", as described in the Materials and Methods. The red and green colors, according to shading, show an increase and decrease in gene expression, respectively, with $D_2O$ treatment compared to $H_2O$ treatment. A characteristic group of genes with increased expression, inhibitors of cell cycle, is surrounded by red dashed lines.
(PDF)

**S14 Fig. Effects of $D_2O$ on expression levels of cellular genes involved in "homologous recombination".** The RNA-seq data shown in S4 Fig were used. Expression levels of each gene were visualized on KEGG pathway map of "homologous recombination", as described in the Materials and Methods. The red and green colors, according to shading, show an increase and decrease in gene expression, respectively, with $D_2O$ treatment compared to $H_2O$ treatment. A characteristic gene with increased expression, an inhibitor of HR repair, is surrounded by red dashed lines.
(PDF)

**S15 Fig. Effects of $D_2O$ on expression levels of cellular genes involved in "nucleotide excision repair".** The RNA-seq data shown in S4 Fig were used. Expression levels of each gene were visualized on a KEGG pathway map of "nucleotide excision repair", as described in the Materials and Methods. The red and green colors, according to shading, show increased and decreased gene expression, respectively, with $D_2O$ treatment compared to $H_2O$ treatment.
(PDF)

**S16 Fig. Effects of $D_2O$ on expression levels of cellular genes involved in "base excision repair".** The RNA-seq data shown in S4 Fig were used. Expression levels of each gene were visualized on a KEGG pathway map of "base excision repair", as described in the Materials and Methods. The red and green colors, according to shading, show increased and decreased gene expression, respectively, with $D_2O$ treatment compared to $H_2O$ treatment. In the upper left part of this figure, please replace Unsaturates with Unsaturated, and then underneath that part, ribosylaion with ribosylation. Underneath that, please replace ricruitment with recruitment.
(PDF)

**S17 Fig. Effects of $D_2O$ on expression levels of cellular genes involved in "mismatch repair".** The RNA-seq data shown in S4 Fig were used. Expression levels of each gene were visualized on a KEGG pathway map of "mismatch repair", as described in the Materials and Methods. The red and green colors, according to shading, show increased and decreased gene expression, respectively, with $D_2O$ treatment compared to $H_2O$ treatment.
(PDF)

**S18 Fig. Effects of D$_2$O on expression levels of cellular genes involved in "Fanconi anemia pathway".** The RNA-seq data shown in S4 Fig were used. Expression levels of each gene were visualized on a KEGG pathway map of "Fanconi anemia pathway", as described in the Materials and Methods. The red and green colors, according to shading, show increased and decreased gene expression, respectively, with D$_2$O treatment compared to H$_2$O treatment.
(PDF)

**S1 Data. The obtained read counts data from RNA-seq experiments.** The CSV file of the S1 Data used for the iDEP96 analysis. S5–S17 Figs were obtained with S1 Data.
(XLSX)

**S2 Data. The experimental design file for the iDEP96 analysis.** The CSV file of S2 Data used for the iDEP96 analysis with the CSV file of S1 Data.
(XLSX)

**S1 Raw image. The original blot and gel images contained in the manuscript's main figures and supplemental figures.** The cropped areas used in the manuscript's main figures and supplemental figures are surrounded by red lines.
(PDF)

## Acknowledgments

We thank Dr. M. Jasin (Memorial Sloan-Kettering Cancer Center, NYC, USA) for the DR-GFP assay materials. We deeply appreciate Dr. S. Hirayama (Emeritus Professor, Kyoto Institute of Technology, Kyoto, Japan) and Dr. H. Kitoh-Nishioka (Kindai University, Osaka, Japan) for helpful instructions and discussions about quantum tunneling and kinetic isotope effects. We also thank Dr. H. Kurumizaka and his laboratory members (The University of Tokyo, Tokyo, Japan) for providing purified histone proteins. This research was supported by Research Support Project for Life Science and Drug Discovery (Basis for Supporting Innovative Drug Discovery and Life Science Research (BINDS)) from AMED under Grant Number JP23ama121009 (H. Kurumizaka).

## Author Contributions

**Conceptualization:** Takeshi Yasuda.

**Funding acquisition:** Takeshi Yasuda, Takaya Gotoh, Wataru Kagawa, Kaoru Sugasawa, Katsushi Tajima.

**Investigation:** Takeshi Yasuda, Nakako Nakajima, Tomoo Ogi, Tomoko Yanaka, Izumi Tanaka, Takaya Gotoh.

**Project administration:** Takeshi Yasuda.

**Resources:** Takeshi Yasuda, Wataru Kagawa, Kaoru Sugasawa.

**Visualization:** Takeshi Yasuda, Nakako Nakajima, Tomoko Yanaka.

**Writing – original draft:** Takeshi Yasuda.

**Writing – review & editing:** Tomoo Ogi, Wataru Kagawa, Kaoru Sugasawa, Katsushi Tajima.

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
