## [Decision Letter · Decision Letter 0]

24 Jul 2024

PONE-D-24-22759Heavy water inhibits DNA double-strand break repairs and disturbs cellular transcription,presumably via quantum-level mechanisms of kinetic isotope effects on hydrolytic enzyme reactionsPLOS ONE

Dear Dr. Yasuda,

Thank you for submitting your manuscript to PLOS ONE. After careful consideration, we feel that it has merit but does not fully meet PLOS ONE’s publication criteria as it currently stands. Therefore, we invite you to submit a revised version of the manuscript that addresses the points raised during the review process.

We look forward to receiving your revised manuscript.

Kind regards,

A Ganesan

Academic Editor

PLOS ONE

Journal Requirements:

We thank Dr. M. Jasin (Memorial Sloan-Kettering Cancer Center, NYC, USA) for the DR-GFP assay materials. We deeply appreciate Dr. S. Hirayama (Emeritus Professor, Kyoto Institute of Technology, Kyoto, Japan) and Dr. H. Kitoh-Nishioka (Kindai University, Osaka, Japan) for helpful instructions and discussions about quantum tunneling and kinetic isotope effects, and Dr. H. Kurumizaka and his laboratory members (The University of Tokyo, Tokyo, Japan) for purified histone proteins. This research was supported by Research Support Project for Life Science and Drug Discovery (Basis for Supporting Innovative Drug Discovery and Life Science Research (BINDS)) from AMED under Grant Number JP22ama121009 (H. Kurumizaka).

This study was funded by JSPS KAKENHI Grant Numbers 20K12177 (to T. Yasuda), 20K08071 (to K.T. and T. Yasuda), 22H03743 (to W.K. and T. Yasuda), and 22K08414 (to T.G. and T. Yasuda). This study was also funded by the joint research program of the Biosignal Research Center, Kobe University, Grant Numbers 281004 (to T. Yasuda and K.S.), and 201003 (to T. Yasuda and K.S.). The funders had no role in the study design, data collection, interpretation, or decision to submit the work for publication.

Reviewers' comments:

Reviewer's Responses to Questions

**Comments to the Author**

1. Is the manuscript technically sound, and do the data support the conclusions?

Reviewer #1: Yes

2. Has the statistical analysis been performed appropriately and rigorously? 

Reviewer #1: Yes

3. Have the authors made all data underlying the findings in their manuscript fully available?

Reviewer #1: Yes

4. Is the manuscript presented in an intelligible fashion and written in standard English?

Reviewer #1: Yes

5. Review Comments to the Author

**Reviewer #1: **General comments to the paper entitled: Heavy water inhibits DNA double-strand break repairs and disturbs cellular transcription, presumably via quantum-level mechanisms of kinetic isotope effects on hydrolytic enzyme reactions

It is well known that the heavy hydrogen, the deuterium, is toxic in high concentrations. The author aimed to reveal the mechanism of deuterium, which is responsible for its toxic effect. To do that, the kinetic isotope effect on different enzymatic reactions was investigated in in vitro and in vivo systems. The study nicely presents the effect on several enzymatic reactions, such as deacetylation, DNA cleavage, and protein cleavage. This excellent study shows the importance of quantum mechanisms fundamentally determining the basis of life.

The critical phenomenon of quantum tunneling involves comparing the behavior of the two stable hydrogen isotopes in the chemical reaction. Due to the 100% mass difference, the quantum tunneling effect is well documented in enzymes with hydrogen transfer.

Congratulations to the authors on their excellent work relating to all the studies applying D2O in a wide range of concentrations.

However, the last sentence of the “Quantum-level effects on life phenomena” paragraph suggests the authors missed something relating to deuterium research. The “D exists as hydrogen, which is essentially negligible on the Earth…” is in line with the concept accepted for 60 years after the discovery of D; we do not need to count on living organisms.

Over 30 years ago, research started to investigate the biological effect of deuterium depletion, specifically the effect of D concentration below the natural level. Today, it is widely accepted that naturally occurring D has a central role in regulating cell growth, tumor development, and metabolism.

I strongly encourage the authors to consider adding one more paragraph to the manuscript, citing papers on deuterium-depleted water (DDW) research. The tunneling effect is present in the cells even at a natural concentration of D. The potential impact of deuterium depletion on cell growth, gene expression, and metabolism suggests the tunneling effect's importance is even higher than the present manuscript suggests. Your research has the potential to give a clear explanation of the tunneling effect for the observed findings with DDW.

6. PLOS authors have the option to publish the peer review history of their article (what does this mean?). If published, this will include your full peer review and any attached files.

Reviewer #1: No

---

## [Author Response · Author response to Decision Letter 0]

15 Aug 2024

Dear Editor:

We are grateful to the academic editor and the reviewers for carefully reading our manuscript, and providing valuable comments that have helped us improve it. We have considered all of their comments, and incorporated the comments as much as possible in the revised version of our paper.　The revised parts of the manuscript are highlighted in red.

We look forward to hearing from you regarding our submission and to responding to any future questions and comments you may have.

Sincerely, 

Takeshi Yasuda

Institute for Quantum Life Science (iQLS), National Institutes for Quantum Science and Technology (QST), 4-9-1 Anagawa, Inage-ku, Chiba 263-8555, Japan

Tel: +81 43 206 3117

Fax: +81 43 206 4094

E-mail: yasuda.takeshi@qst.go.jp

Response to the comments from the academic editor

Journal Requirements:

Thank you for your kind comment. The following corrections have been made in accordance with PLOS ONE's style requirements.

The previous manuscript included a Full title and a Short title, but the Short title was removed in the revised manuscript.

The ZIP codes and abbreviations were removed from the Affiliations.

The information on the corresponding author was revised to include only the email address. As for the initials of the corresponding author after the e-mail address, we chose "Takeshi Y" to distinguish Takeshi Yasuda and Tomoko Yanaka, since they have the same TY initials.

All major sections (Abstract, Introduction, Materials and methods, Results, Discussion, etc.) were changed to 18pt font.

The PDF files of the main figure were changed to Tif files and renamed as "Fig1.tif", etc.

Supporting information files are uploaded separately as individual files. 

The Supporting information file names were changed to "S1 Text.pdf", "S1 Fig.pdf", etc.

The "Materials and Methods" section has been moved to the next section after the "Introduction". The numbering of the references has been corrected accordingly.

We left the information about the two preprints on which this paper is based in the manuscript, because we could not determine whether it should be removed.

Finally, we have replaced the KEGG permission file for PLOS Biology with the KEGG permission file for PLOS ONE.

We thank Dr. M. Jasin (Memorial Sloan-Kettering Cancer Center, NYC, USA) for the DR-GFP assay materials. We deeply appreciate Dr. S. Hirayama (Emeritus Professor, Kyoto Institute of Technology, Kyoto, Japan) and Dr. H. Kitoh-Nishioka (Kindai University, Osaka, Japan) for helpful instructions and discussions about quantum tunneling and kinetic isotope effects, and Dr. H. Kurumizaka and his laboratory members (The University of Tokyo, Tokyo, Japan) for purified histone proteins. This research was supported by Research Support Project for Life Science and Drug Discovery (Basis for Supporting Innovative Drug Discovery and Life Science Research (BINDS)) from AMED under Grant Number JP22ama121009 (H. Kurumizaka).

This study was funded by JSPS KAKENHI Grant Numbers 20K12177 (to T. Yasuda), 20K08071 (to K.T. and T. Yasuda), 22H03743 (to W.K. and T. Yasuda), and 22K08414 (to T.G. and T. Yasuda). This study was also funded by the joint research program of the Biosignal Research Center, Kobe University, Grant Numbers 281004 (to T. Yasuda and K.S.), and 201003 (to T. Yasuda and K.S.). The funders had no role in the study design, data collection, interpretation, or decision to submit the work for publication.

Thank you for pointing out the corrections. The histone proteins were provided by Dr. Kurumizaka through the support of the Research Support Project for Life Science and Drug Discovery (Basis for Supporting Innovative Drug Discovery and Life Science Research (BINDS)). The Grant (Grant Number JP22ama121009) was obtained by Dr. Kurumizaka, not by us. According to a statement from the BINDS office, BINDS-supported research must be mentioned in the acknowledgements as follows: "This research was supported by Research Support Project for Life Science and Drug Discovery (Basis for Supporting Innovative Drug Discovery and Life Science Research (BINDS)) from AMED under Grant Number JP22ama121xxx". Therefore, we have included this sentence in the Acknowledgments Section instead of the Funding Statement. In consultation with the BINDS office, this sentence was moved to the Funding Statement. The Grant Number JP22ama121009 was changed to Grant Number JP23ama121009 at the request of the Kurumizaka Laboratory. Instead, in the Acknowledgments section, we removed the grant number information and revised the following sentence: "We also thank Dr. H. Kurumizaka and his laboratory members (The University of Tokyo, Tokyo, Japan) for providing purified histone proteins via Research Support Project for Life Science and Drug Discovery (Basis for Supporting Innovative Drug Discovery and Life Science Research (BINDS)) from AMED." We removed any funding-related text from the manuscript, and updated our Funding Statement as follows: "This study was funded by JSPS KAKENHI Grant Numbers 20K12177 (to T. Yasuda), 20K08071 (to K.T. and T. Yasuda), 22H03743 (to W.K. and T. Yasuda), and 22K08414 (to T.G. and T. Yasuda). This study was also funded by the joint research program of the Biosignal Research Center, Kobe University, Grant Numbers 281004 (to T. Yasuda and K.S.), and 201003 (to T. Yasuda and K.S.). This research was supported by Research Support Project for Life Science and Drug Discovery (Basis for Supporting Innovative Drug Discovery and Life Science Research (BINDS)) from AMED under Grant Number JP23ama121009 (H. Kurumizaka). The funders had no role in the study design, data collection, interpretation, or decision to submit the work for publication."

Thank you for pointing out the corrections. We added our original blot/gel image data (S1 Raw images.pdf) in the Supporting Information. Accordingly, the following sentences have been added to the legends of the Supporting information of the manuscript: "S1 Raw images. The original blot and gel images contained in the manuscript’s main figures and supplemental figures. The cropped areas used in the manuscript’s main figures and supplemental figures are surrounded by red lines."

Thank you for your kind comment. We have verified that the reference list is complete and correct. Also, we did not cite any RETRACTED papers.

Response to the comments from the reviewer #1

However, the last sentence of the “Quantum-level effects on life phenomena” paragraph suggests the authors missed something relating to deuterium research. The “D exists as hydrogen, which is essentially negligible on the Earth…” is in line with the concept accepted for 60 years after the discovery of D; we do not need to count on living organisms.

Thank you for your kind comments. Your insights have highlighted the importance of the effects of naturally occurring deuterium concentrations. According to your comments, we have removed the following phrase: "which is essentially negligible on the Earth."

Over 30 years ago, research started to investigate the biological effect of deuterium depletion, specifically the effect of D concentration below the natural level. Today, it is widely accepted that naturally occurring D has a central role in regulating cell growth, tumor development, and metabolism.

I strongly encourage the authors to consider adding one more paragraph to the manuscript, citing papers on deuterium-depleted water (DDW) research. The tunneling effect is present in the cells even at a natural concentration of D. The potential impact of deuterium depletion on cell growth, gene expression, and metabolism suggests the tunneling effect's importance is even higher than the present manuscript suggests. Your research has the potential to give a clear explanation of the tunneling effect for the observed findings with DDW.

Thank you for your understanding and appreciation of our research. We are particularly grateful that you pointed out the relevance of our results concerning the biological effects of deuterium-depleted water, which we had not previously considered. Your comments have greatly enhanced the significance of our findings. In line with your suggestions, we have added the following text to the end of the Discussion, citing the relevant papers (references 98-104) on DDW: "While the biological effects of high concentrations of heavy water are strong, there are also notable effects when the deuterium content in naturally occurring water is depleted [98-104]. Compared to normal water, deuterium-depleted water (DDW) reduces the growth rate of mammalian cells [98-100]. The inhibitory effect of DDW on cancer cell proliferation has attracted keen attention, from the perspective of cancer treatment [99-102]. DDW also alters gene expression [101, 104] and the amounts of intracellular metabolic products [102, 103]. Thus, like heavy water at high concentrations, DDW affects the same biological phenomena such as cell proliferation and transcription. Therefore, our study of quantum effects may provide a clear explanation for the observed biological effects of DDW, which could also result from quantum-level isotope effects on the numerous hydrolytic enzyme reactions within cells. Based on our findings and those of previous studies, it can be inferred that quantum-level mechanisms, including quantum tunneling, in hydrolytic enzyme reactions must have important effects on the fundamental life phenomena of organisms living on Earth."

---

## [Editor Report · Decision Letter 1]

16 Aug 2024

Heavy water inhibits DNA double-strand break repairs and disturbs cellular transcription,presumably via quantum-level mechanisms of kinetic isotope effects on hydrolytic enzyme reactions

PONE-D-24-22759R1

Dear Dr. Yasuda,

We’re pleased to inform you that your manuscript has been judged scientifically suitable for publication and will be formally accepted for publication once it meets all outstanding technical requirements.

Kind regards,

A Ganesan

Academic Editor

PLOS ONE
---

## [Editor Report · Acceptance letter]

10 Sep 2024

PONE-D-24-22759R1 

PLOS ONE

Dear Dr. Yasuda, 

I'm pleased to inform you that your manuscript has been deemed suitable for publication in PLOS ONE. Congratulations! Your manuscript is now being handed over to our production team.

Kind regards, 

on behalf of

Prof. A Ganesan 

Academic Editor

PLOS ONE